# From Near-Sensor to In-Sensor: A State-of-the-Art Review of Embedded AI Vision Systems

**DOI:** 10.3390/s24165446

**Published:** 2024-08-22

**Authors:** William Fabre, Karim Haroun, Vincent Lorrain, Maria Lepecq, Gilles Sicard

**Affiliations:** Université Paris-Saclay, CEA, List, F-91120 Palaiseau, France; karim.haroun@cea.fr (K.H.); vincent.lorrain@cea.fr (V.L.); maria.lepecq@cea.fr (M.L.); gilles.sicard@cea.fr (G.S.)

**Keywords:** vision systems, sensor processing, neural networks, real-time processing, embedded systems, AI vision, energy efficiency, embedded vision systems

## Abstract

In modern cyber-physical systems, the integration of AI into vision pipelines is now a standard practice for applications ranging from autonomous vehicles to mobile devices. Traditional AI integration often relies on cloud-based processing, which faces challenges such as data access bottlenecks, increased latency, and high power consumption. This article reviews embedded AI vision systems, examining the diverse landscape of near-sensor and in-sensor processing architectures that incorporate convolutional neural networks. We begin with a comprehensive analysis of the critical characteristics and metrics that define the performance of AI-integrated vision systems. These include sensor resolution, frame rate, data bandwidth, computational throughput, latency, power efficiency, and overall system scalability. Understanding these metrics provides a foundation for evaluating how different embedded processing architectures impact the entire vision pipeline, from image capture to AI inference. Our analysis delves into near-sensor systems that leverage dedicated hardware accelerators and commercially available components to efficiently process data close to their source, minimizing data transfer overhead and latency. These systems offer a balance between flexibility and performance, allowing for real-time processing in constrained environments. In addition, we explore in-sensor processing solutions that integrate computational capabilities directly into the sensor. This approach addresses the rigorous demand constraints of embedded applications by significantly reducing data movement and power consumption while also enabling in-sensor feature extraction, pre-processing, and CNN inference. By comparing these approaches, we identify trade-offs related to flexibility, power consumption, and computational performance. Ultimately, this article provides insights into the evolving landscape of embedded AI vision systems and suggests new research directions for the development of next-generation machine vision systems.

## 1. Introduction

In the age of artificial intelligence, computer vision leverages edge computing [1] to analyze complex visual data in real time, transforming machine–environment interactions with unprecedented precision [2].

Vision systems capture images and transfer them to processing units, where the raw visual data are transformed into usable information. This process is used in applications such as autonomous navigation, augmented reality, mobile phones, and drones, where the rapid analysis of visual data enables fluid interactions and real-time decisions.

Before the advent of AI-based techniques, image processing relied on custom algorithms that required considerable manual tuning and expertise. Methods such as SIFT (Scale-Invariant Feature Transform) [3] and SURF (Speeded-Up Robust Features) [4] were used for feature detection and object recognition but required precise parameter adjustments and struggled to generalize to diverse datasets. HOG (Histogram of Oriented Gradients) [5] was widely used for pedestrian detection due to its ability to capture gradients but was sensitive to variations in lighting and perspective. These traditional algorithms were time-consuming to develop and often failed to adapt to new environments, making them labor-intensive and limiting their effectiveness in dynamic scenarios.

The limitations of traditional image processing methods, which heavily rely on manual feature engineering and lack of adaptability, have paved the way for more sophisticated approaches. Approaches based on deep neural networks (DNNs), particularly convolutional neural networks (CNNs), have emerged as groundbreaking solutions [6,7,8]. This shift is largely due to advances in GPU technology, data storage, and the availability of large datasets [9,10]. Unlike customized algorithms, CNNs automatically learn hierarchical features from raw data. This ability allows them to excel at complex tasks such as object detection, recognition, and segmentation [11,12,13]. This capability allows CNNs to generalize to diverse datasets and adapt to dynamic environments, thus overcoming the limitations of previous methods.

CNNs are generally structured into two main components: the backbone and the head. The backbone is responsible for extracting spatial features from images using convolutional layers, while the head often uses a multilayer perceptron (MLP) for tasks such as image classification [14]. This architecture enables CNNs to efficiently process and analyze image data, making them an essential component of modern computer vision applications.

While recent advances in Pure-MLP networks [15,16] and attention-based models, such as Transformers [17,18] demonstrate significant potential in processing complex vision tasks, they are often accompanied by substantial computational requirements. For example, the Vision Transformer (ViT-L/16) requires approximately 190 billion FLOPs and 307 million parameters [17], illustrating the high resource requirements. Even highly optimized models like the Swin Transformer require 4.5 billion FLOPs [19], putting them on par with the most computationally intensive CNN models.

Recent Pure-MLP models, such as CycleMLP, attempt to bridge the gap between performance and efficiency but still require around 2.1 billion FLOPs and 15 million parameters [20]. Although this is an improvement over earlier MLP architectures, the computational costs remain significant compared to embedded CNNs, especially in scenarios where power and computational resources are limited. The gMLP model further emphasizes these challenges, with 15 billion FLOPs and 73 million parameters required to achieve competitive accuracy [21], making them less practical for deployment on edge devices.

Although new models of vision processing, such as Pure-MLP networks and Transformers, have emerged, CNNs maintain a prominent position in artificial intelligence solutions due to their effectiveness and extensive refinement over the years. These neural networks have consistently demonstrated strong performance across various applications, making them a preferred choice for many AI-based vision systems. As the demand for complex computing tasks grows, industries increasingly rely on cloud infrastructure to handle data processing and analysis. However, this shift introduces significant challenges in the context of cloud computing that require further innovation in designing and deploying AI-driven vision systems:Sensors generate massive amounts of data, which can lead to processing and communication bottlenecks in cloud computing.Real-time processing requirements often exceed the latency capabilities of cloud computing.Computation and communication with or in the cloud, and specifically access to the data generated by computations, require high energy consumption [22].

To address these challenges, efforts are being made to integrate complex processing directly into embedded vision systems, enabling autonomous image capture as well as analysis. This approach reduces the need for communication with the cloud and the use of remote cloud computing resources, optimizing overall energy consumption and processing latency.

By integrating advanced processing capabilities into these systems, AI-embedded vision systems can achieve greater autonomy and efficiency. An embedded vision system typically comprises a camera for initial data capture and rudimentary image processing, combined with a processing unit that refines image quality and performs advanced analysis, ultimately transmitting the analyzed visual information (as illustrated in Figure 1-(1, …, n + 2)). This integration allows systems to perform CNN processing close to or within the vision system itself, reducing dependence on external resources and improving real-time processing capabilities.

In this study, our objective is to compare AI-embedded vision systems operating in the visible spectrum that integrate CNN processing. We evaluate these systems in terms of their power consumption, latency, and performance metrics, providing an overview of the current design landscape and its categories.

### 1.1. Embedded AI Vision Systems Categories

We categorize embedded vision systems into two types: flexible embedded AI vision systems and ultra-embedded AI vision systems.

Flexible embedded AI vision systems offer latency advantages over cloud-based systems by minimizing data transmission requirements. While not inherently focused on low-power consumption, these systems provide high flexibility to accommodate various standard application software [26,27,28]. Flexible embedded AI vision systems are capable of handling complex CNN processing tasks. These systems require an energy budget ranging from 1 W to 300 W for capturing and processing images. As such, they are ideally suited for autonomous vehicles that can support embedded high energy demands [29].

Ultra-embedded AI vision systems are tailored for low-power tasks near or within the sensor. Ideal for small-sized, battery-dependent devices, these systems excel in environments like smartphones and small drones, where energy consumption for vision capture and processing remains under 1 W [30].

### 1.2. Design Space: From Near-Sensor to In-Sensor

The main challenge for embedded AI vision systems is balancing the trade-off between the solution’s flexibility and its power consumption, as demonstrated in Figure 2. These systems range from cloud-based solutions, which are highly flexible but consume significant energy, to ultra-low-power devices that are energy-efficient but offer limited flexibility.

To effectively integrate CNNs into these vision systems, research is primarily focused on two approaches: processing near the sensor (near-sensor) and directly inside the sensor (in-sensor). These strategies are crucial for optimizing performance and energy use, enabling the more efficient deployment of AI capabilities.

#### 1.2.1. Near-Sensor AI Vision Systems

Near-sensor AI vision systems use separate circuits for each function of the embedded vision system. Although the system might be composed of some dedicated hardware, it usually integrates Components On The Shelf (COTSs), which facilitate the rapid development of the entire vision system.

In these systems, different processing units can be used, such as graphics processing units (GPUs), central processing units (CPUs), tensor processing units (TPUs), and neural processing units [8] (NPUs), depending on factors such as system size, energy efficiency, and flexibility. The choice of these components depends on the level of specialization required by the AI application.

This design flexibility, combined with the targeted approach to component selection, means that these embedded vision systems can be developed for a wide range of applications, with a wide range of possible sizes and power consumption, within short production times.

#### 1.2.2. In-Sensor AI Vision Systems

In-sensor AI vision systems offer a higher level of integration by integrating various processing functionalities directly into the sensor. These systems offer significant advantages in terms of parallelism for both data access and computation.

These systems are divided into two categories: Optimized NPU-based Processing (ONP) and Highly Parallel and Distributed Processing (HPDP):

ONP: This category includes 3D IC vision systems that leverage 3D integration technologies [52] to address the bottleneck between image capture and CNN processing [31]. Three-dimensional IC vision systems specialize CMOS layers for specific functions, providing further opportunities to integrate CNN processing as close to the image sensor as possible. The main challenges and opportunities of this technology involve rethinking data access, partitioning the computational architecture, and CMOS layer specialization for specific tasks such as image acquisition or CNN processing.

The HPDP category involves two generic processing approaches that offer innovative AI capabilities through highly parallel and distributed processing.

Two-dimensional in-pixel processing: Two-dimensional in-pixel processing vision systems [53,54] consist of photosensitive matrices that integrate processing elements into individual pixels or groups of pixels, allowing for parallel spatial processing across the pixel array to implement convolution operations. The main challenges for these systems include efficiently distributing calculations and CNN weights on its array, optimizing data access, and organizing data movements with a shared memory to ensure efficient processing.

Three-dimensional macropixel array: This approach uses 3D IC technology to construct three-dimensional pixel structures [55] called macropixels. This approach enables enhanced data access and highly parallel computational capabilities by leveraging 3D hardwired communications and distributing Analog-to-Digital Converters (ADCs) across the entire circuit [23].

Other approaches, such as in-memory computing (IMC), explore the potential of processing directly within memory arrays to achieve high performance with low energy consumption [56,57]. Although promising, these are not discussed further in this study, as IMC represents a distinct computational approach that diverges from the traditional architecture of embedded AI vision systems covered here.

In this article, we discuss the different types of vision system architectures that incorporate neural network capabilities and the trade-offs considered in each category. The rest of the article is organized as follows: In Section 2, we discuss the fundamental metrics and characteristics of AI vision systems. This detailed examination provides the essential terminology and concepts that will be used throughout the subsequent sections, ensuring a consistent and clear understanding of the performance and operational principles of these systems. Section 3 provides an overview of AI vision systems that rely on dedicated hardware and COTS components to perform near-sensor processing. In Section 4, we discuss in-sensor processing vision systems with AI capabilities. Finally, in Section 5, we conclude the article and propose potential directions for future research.

## 2. Metrics and Characteristics

This section delves into the metrics and characteristics of AI-integrated computer vision systems, offering a comprehensive analysis from image capture to CNN processing. We will explore key performance metrics and design trade-offs that are essential for evaluating these systems following the sequence in Figure 1.

### 2.1. Image Acquisition Characteristics

In this subsection, we explore the initial stage of the computer vision system pipeline, starting with image acquisition (as shown in Figure 1-(1)). This process converts the analog signal into digital data for further processing. The key parameters of image acquisition are acquisition resolution and processing resolution. They determine the precision with which a vision system can capture and analyze visual data.

#### 2.1.1. Capture Resolution

Vision systems’ capture resolution refers to the number of pixels in the sensor, which is closely linked to the sensor’s size and pixel dimensions. Higher resolutions provide richer information, crucial for identifying detailed features in captured images, at the expense of increased data volume. This greater volume can lead to heightened demands on processing power and storage, potentially increasing system latency and power consumption.

#### 2.1.2. Processing Resolution

Processing resolution is determined by the specifications of the CNN used for image analysis and is often lower than the capture resolution. Adjustments such as resizing, cropping, and other pre-processing steps are required to adapt the captured image to the appropriate dimensions to be processed by the CNN. These modifications are necessary to manage the computational and memory requirements of the CNNs, which tend to increase with the size of the input image [58,59].

Once captured, the image is transferred to the next stage in the vision system pipeline, where pixel-level processing takes place. This stage converts the raw signal data into a digital representation that can be accurately interpreted by the CNN processing pipeline.

### 2.2. Pixel-Level Processing

When inferring a CNN, pixel-level processing (Figure 1-(2)) is used to ensure optimal accuracy and robustness of the CNN model to artifacts in the image data. This pre-processing is generally performed in an Image Signal Processor (ISP), which is a dedicated hardware component designed to rapidly process and enhance digital images. It can also be highly integrated into the vision system [32].

#### 2.2.1. Image Signal Processor

Providing a bridge between raw image data and CNN input requirements, the ISP applies essential pre-processing to optimize, correct, and adapt captured images in real time. This ensures that the images are ideally conditioned for subsequent analysis by CNNs. Another solution would be to perform no pre-processing at all [33] to save latency and energy at the expense of the performance of the CNN application.

#### 2.2.2. Pre-Processing

Commonly used pre-processing techniques include the normalization of pixel values to reduce variations in brightness and contrast [60], resizing to adjust the capture resolution of the camera to the processing resolution [31], and ROI detection [32] to select particular regions of interest, such as facial detection [61].

To illustrate the benefits of pre-processing, consider the technique of binning. Binning is a method used to reduce the amount of data that need to be processed and transmitted by combining multiple pixel values into one. This reduces the resolution and, consequently, the bandwidth requirements. For example, a camera with an original resolution of 4000×3000 pixels, or 12 MPs, can have its pixel count reduced by a factor of four using 2 × 2 binning:(1)OriginalPixelCount=4000×3000=12,000,000pixels=12MP

This results in a binned resolution of 2000×1500 pixels, or 3 MPs:(2)$$\mathrm{Binned}\;\mathrm{Pixel}\;\mathrm{Count}=\frac{4000\times 3000}{2\times 2}=2000\times 1500=3,000,000\;\mathrm{pixels}=3\;\mathrm{MP}$$

Binning illustrates how pre-processing can effectively reduce the amount of data to be transmitted by lowering resolution and bandwidth requirements. This process becomes particularly important in the transition to digital processing, where efficient bandwidth management is needed to avoid bottlenecks. This management via appropriate interconnections is essential in the next phase of the vision pipeline as efficient communication between components, such as the ISP, AI processing unit, and memory, plays an important role in maintaining system performance.

### 2.3. Bandwidth and Interconnect

Data access poses significant challenges when implementing CNNs within embedded systems. Two key bandwidths must be managed: the analog-to-digital bandwidth, which transfers pixel signals from the camera’s readout to the pixel-level processing unit (as shown in Figure 1-(3)), and the digital interconnect bandwidth, which facilitates communication between the pixel-level processing unit, memory, and CNN accelerator (Figure 1-(4, 5, 6, …, n, n + 1, n + 2). The effective management of these bandwidths is essential to reduce latency and improve overall system performance.

#### 2.3.1. Analog-to-Digital Bandwidth

This bandwidth primarily concerns the first layer of the CNN and involves the pre-processing stage to reduce the amount of data via ISP. Often, there is a direct connection between the image sensor and the ISP, with internal storage capacity within the ISP. For example, consider a 12 MP sensor with 10-bit pixels operating at 30 fps. The raw data transfer bandwidth requirement is expressed as follows:(3)Bandwidth=4000×3000pixels×10bits/pixel×30fps=3.6Gbps

Transferring all these data to a cloud processor can create bottlenecks in an embedded device. Utilizing near-sensor processors can significantly reduce the required data transfer and power consumption compared to transferring all the data via Wi-Fi [34,35,36].

Other, more integrated approaches are based on the distribution of the pixel readout. We consider different, more ad hoc configurations. First is a group of 16 processors for a group of pixels, as found in stacked 3D systems HPDP [62] (for example, 16 × 16 pixels):(4)$$\mathrm{Number}\;\mathrm{of}\;\mathrm{groups}=\frac{4000\times 3000}{16\times 16}=46,875$$
(5)Bandwidthpergroup=256pixels/group×10bits/pixel×30fps=0.0768Gbps/group

Another approach could involve an image sensor column-parallel ADC readout [31,63,64].
(6)Bandwidthpercolumn=3000pixels×10bits/pixel×30fps=0.9Gbps/column

With 4000 columns processed in parallel, the total bandwidth remains 3.6 Gbps, but the load per column is reduced.

Finally, for a processor per pixel found in 2D HPDP systems [53],
(7)Bandwidthperpixel=10bits×30fps=300bitspersecond(bps)perpixel

These different approaches distribute the reading task by groups or columns, reducing the bandwidth per processing unit but requiring more complex integration. While a processor per pixel maximizes parallelism and minimizes latency, it significantly increases the integration complexity, thesurface of the readout elements, and power consumption during reading [37].

#### 2.3.2. Digital Interconnect Bandwidth

Embedded AI vision systems process vast volumes of data, particularly in convolutional layers of CNN, which require efficient dataflow management for input feature maps (ifmaps), kernels, partial sums (psums), and output feature maps (ofmaps). The choice of communication bus directly influences overall system performance, power consumption, and physical size. External serial buses, such as PCIe (Peripheral Component Interconnect Express), offer high bandwidths of up to 64 GB/s with PCIe 5.0 [65,66]. Serial buses transmit data bit by bit over multiple lanes, enabling fast transmission over long distances. PCIe is designed to exchange data between system components, such as processors and graphics cards, which are ideal for large-scale parallel calculations and massive data processing. However, they have high power consumption and require a significant physical footprint for their connectors and cables, making them less suitable for compact, low-power systems.

MIPI CSI-2 (Camera Serial Interface 2) is another external serial bus used for image data transmission. It is designed for the efficient transfer of high-definition video streams in embedded vision applications. MIPI CSI-2 offers a bandwidth of up to 2.5 Gbps per channel, with a typical configuration of 4 to 8 channels, enabling a total bandwidth of 10 to 20 Gbps to be achieved [31,67,68,69]. This protocol is optimized for mobile devices, drones, and other IoT devices, where energy efficiency and performance are critical.

Internal buses, such as AMBA AXI (Advanced eXtensible Interface) [31,70], use a parallel bus architecture that enables multiple bits to be transmitted simultaneously over multiple lines, offering moderate bandwidths of 1 to 20 Gbps. This approach reduces latency and enables efficient data management within the chip while minimizing the physical footprint and power consumption. Lightweight protocols such as I2C [69,71] and SPI [68,72], which are also based on a serial bus architecture, offer more limited bandwidth, up to 3.4 Mbps for I2C and 60 Mbps for SPI. These protocols are simple to implement and cost-effective, ideal for sensor configuration and control in environments where space is limited and high bandwidth is not required.

Although bandwidth enables rapid data movement, the effectiveness of embedded AI vision systems is determined by the appropriate memory size and efficient data management. Insufficient memory leads to performance bottlenecks, which hinder the system’s ability to handle large datasets and complex computations. Proper sizing and managing memory allows for optimal data processing and improved system responsiveness.

### 2.4. Memory Resources

Sufficient memory is essential for storing the CNN model and facilitating the smooth running of system configurations. It influences many operational aspects, such as model loading speed (as shown in Figure 1-(5)), data throughput at run-time, and the ability to handle large datasets or complex network architectures without compromising performance. In vision systems using CNN processing, one of the main challenges is to effectively manage the storage of psums and ofmaps (as shown in Figure 1-(6), …, (n)). The data flow strategy and scheduling of computational loops can lead to excessive accumulation of partial sums if they are not carefully managed, particularly if the computation of a feature map is not completed before proceeding. It is therefore necessary to allocate dedicated memory for temporary storage.

Another important consideration is the need to store the entire ofmap of the most important layer in terms of output size before moving on to the next layer, particularly in architectures that do not support multiple layers simultaneously on the AI processing unit. The need to store the ofmap of this layer becomes a critical factor in determining the size of memory required.

The poor memory management of psums and ofmaps can have a significant impact on processing speed and increase latency. System memory sizing for a target CNN class is therefore essential for efficient model execution, enabling large models to be loaded, processed, and executed in real time without compromising performance or accuracy. The choice of memory resources should be guided by dataflow, parallelism capabilities, and the specific requirements of the most demanding layers of the CNN architecture.

Efficient memory sizing optimizes data storage and retrieval and directly affects the choice of CNNs. As we explore the AI processing unit, we first explore CNNs and their progression in response to evolving integrability challenges and various trade-offs.

### 2.5. Convolutional Neural Networks

This subsection explores CNN-based processing, highlighting its progression in response to evolving challenges in embeddability and the diverse trade-offs involved.

A CNN (illustrated in Figure 3a) is a powerful type of neural network for analyzing image data. Its execution is known as the inference process. During the inference process, it uses filters called kernels, which perform multiplication and accumulation operations (MAC), that are fundamental to the extraction of features by the convolution process. Each filter is applied across different channels of the input—such as the color channels in an RGB image—to extract distinct features. The CNN structure includes convolutional layers with filtering kernels and activation functions that introduce non-linearity, allowing the network to learn complex visual patterns from multichannel inputs.

In a CNN, parameters primarily refer to the weights and biases associated with each kernel (illustrated in Figure 3b). During training, conducted through gradient descent [73], the network iteratively adjusts parameters to enhance model performance based on the provided training data. This adjustment involves fine-tuning weights and biases to minimize the error between the predicted outputs of the model and the true labels [74]. The quantity of these parameters can significantly influence both the performance and computational intensity of the model. A higher number of parameters often means greater representational power, which can lead to more precise feature extraction and accuracy. However, it also increases the model’s computational requirements and memory use, which can be critical when deployed at the edge.

### 2.6. Evolution of Embedded Convolutional Neural Networks

Over time, the integration of CNNs into embedded devices has been driven by the need to balance model complexity and performance, leading to various innovations in model design. We focus specifically on how the topology of CNNs—defining the layout and interaction of layers within the backbone and head, as well as the operations performed within these layers—influence performance. We place particular emphasis on accuracy performance [75] as a critical metric for comparison. Accuracy measures the ratio of correct to total predictions and is particularly crucial in image-classification contexts such as the ImageNet [9] dataset, where the top-1 and top-5 error rates provide a clear measure of efficiency.

For the remainder of this subsection, we explore the temporal evolution of integrated convolutional neural networks in order to better understand the different models used in near-sensor and in-sensor-based computer vision systems. This chronological overview shows how each advance contributed to the development of current models, which combine the strengths of earlier designs. In addition, we classify CNN models optimized for embedded vision systems into three distinct groups, each representing a different balance between ease of implementation, computational requirements, number of parameters, and accuracy (as shown in Figure 4):High-Complexity Models (dark gray): These models feature advanced topology and excellent accuracy but require significant layer modifications to adapt to embedded vision systems. These adaptations may include modifying or removing specific components such as Squeeze-and-Excite modules [76] or Swish activation [77] to make better use of complexity-reduction techniques. These models also require potential adaptation in terms of the number of parameters and calculations, which may not be suitable for certain embedded platforms.Adaptable Integration Models (light gray): This category includes designs that achieve high accuracy and maintain a manageable computing footprint. However, these models often require adaptations, such as modifications to certain layers, due to the large number of parameters and the computing power they require.Optimized for Embedded Use (White): These models are designed specifically for embedded systems, with an emphasis on ease of integration and minimal computational requirements. They use simple computation methods, such as ReLU activation, convolution factorization, and depth-separable convolutions [78], and incorporate techniques such as batch normalization [79] directly into the convolutional layers, as demonstrated by EfficientNet-Lite [59]. Although they can sometimes sacrifice accuracy, their low resource requirements make them ideal for resource-constrained environments.

**Figure 4 sensors-24-05446-f004:**
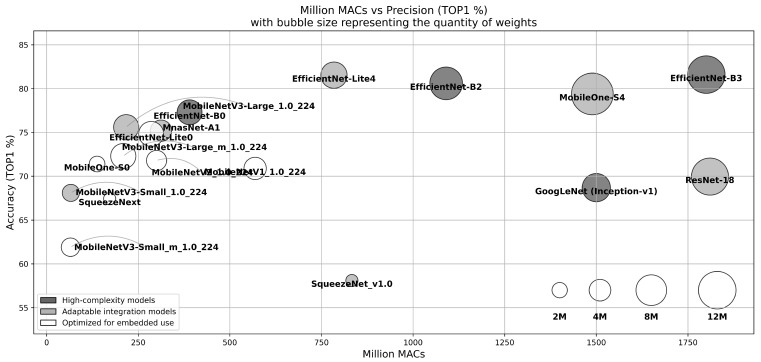
Representation of CNN models for embedded systems, showcasing memory requirements, computational demands, and accuracy [14,58,59,76,80,81,82,83,84,85,86] for classification tasks on ImageNet-1K [9]. Each bubble represents a model, with size indicating memory requirements and position reflecting ImageNet’s computational requirements and model accuracy (inspired by ([75]). Distinctive color patterns in Figure 4 categorize the models by their accuracy and implementation complexity, illustrating the trade-offs between accuracy, number of parameters, MAC requirements, and ease of layer implementation.

The introduction of VGGNet (2014) [87] is the first step towards complexity reductions. VGGNet simplified the topology of AlexNet (2012) [11] by using convolution factorization, allowing it to exclusively use 3 × 3 convolution filters, which resulted in a large reduction in the number of MACs. With approximately 15.5 billion MACs and 138 million parameters, VGGNet falls under the category of High-Complexity Models due to its high computational and memory requirements, even though its layers are simple convolutions.

The arrival of GoogLeNet (2014) [80] marked a turning point in the optimization of embeddability by using inception modules paralleling multiple convolution sizes. By combining convolution factorization and 1 × 1 convolutions to reduce channel dimensionality, GoogLeNet achieved slightly better accuracy than VGGNet, but with approximately 1.5 billion MACs and 6.8 million parameters. However, the need to store temporary data for the different branches of the network demands significant memory, making GoogLeNet a High-Complexity Model.

ResNet [14] introduced residual connections to reduce both computational requirements and memory use. Residual connections are shortcuts that skip one or more layers, improving deep network learning by facilitating gradient backpropagation. An example of ResNet implementation is ResNet-50, with 1.7 billion MACs and 25.6 million parameters, outperforming the VGG16 implementation with a top-5 accuracy of 92.8% on ImageNet. Its reduced computational requirements and memory use classify ResNet as an Adaptable Integration Model due to its more manageable computational footprint. Additionally, there are numerous ResNet variants like ResNet-18 and ResNet-8 that offer easier implementation with various trade-offs in terms of computational demands and accuracy.

SqueezeNet (2016) [81] was one of the first architectures specifically designed for highly constrained systems, introducing the Squeeze-and-Excite (SE) module [76]. This module dynamically adjusts the importance of different features in a layer, allowing the most important features to be strengthened and the less important ones to be reduced, thus improving model accuracy while reducing the number of parameters. SqueezeNet’s lightweight architecture offers accuracy comparable to AlexNet, but with only 860 million MACs and 1.25 million parameters, achieving an accuracy of 58.1%. SqueezeNet is categorized as an Adaptable Integration Model due to its low computational requirements and ease of integration.

SqueezeNext (2018) [83] is the evolution of SqueezeNet, designed to be even more efficient with fewer parameters and lower computational cost. SqueezeNext achieves this through further optimization of the fire modules by simplifying their structure and incorporating bottleneck layers. This results in significantly fewer parameters and MACs while maintaining high performance. For instance, the SqueezeNext 1.0-SqNxt-23 model has 285 million MACs and 1.30 million parameters, while the SqueezeNext 1.0-SqNxt-23v5 model has 348 million MACs and 1.82 million parameters. SqueezeNext uses a similar approach to SqueezeNet with squeeze layers that use 1×1 convolutions to reduce the number of input channels, followed by expand layers that apply 1×1 and 3×3 convolutions. These optimizations allow SqueezeNext to offer an even smaller model size, making it highly suitable for embedded systems with limited resources. SqueezeNext is categorized as Optimized for Embedded Use due to its extremely low computational requirements and ease of integration.

Similarly, the MobileNet family [58,82,86] of networks is targeted for mobile and highly embedded systems. MobileNetV1 (2017) [58] uses depth-wise-separable convolutions introduced by [78]. This method separates the convolution into two layers: a depth-wise (DW) convolution, applying a single filter per input channel, and a point-wise (PW) convolution that combines these outputs, effectively reducing computational complexity and model size. It also uses scaling factors to further lighten the model, considering the size of the input image and the number of parameters. With an input size of 224 × 224, the MobileNetV1_1.0_224 model has 569 million MACs and 4.2 million parameters, while a smaller MobileNetV1_0.5_224 model has 150 million MACs and 1.3 million parameters. Both fit into the Optimized for Embedded Use category.

MobileNetV2 (2018) [86] also uses depth-wise-separable convolutions. It reduces computational requirements and increases accuracy by using inverted residual connections and linear bottlenecks. The process involves an initial 1 × 1 convolution layer to increase the representation (also called expansion), followed by a 3 × 3 depth-wise convolution to extract information from this representation, and a final 1x1 convolution layer (projection) to reduce the data volume. With an input size of 224 × 224, the MobileNetV2_1.0_224 model has 300 million MACs and 3.4 million parameters, while a smaller MobileNetV2_0.5_224 model has 97 million MACs and 1.95 million parameters [88]. Both belong to the Optimized for Embedded Use category.

MobileNetV3 (2019) [82] combined the lessons learned from previous MobileNet versions and MnasNet [84], using Neural Architecture Search techniques [89] to optimize the architecture and SE modules to enhance accuracy. With an input size of 224 × 224, the MobileNetV3-Large_1.0_224 model has 155 million MACs and 4.0 million parameters, while the MobileNetV3-Small_1.0_224 model has 66 million MACs and 2.9 million parameters [90]. MobileNetV3 models are categorized under Adaptable Integration Models due to their advanced topology and lightweight architecture.

EfficientNet (2020) [59] provides a method for coordinately scaling network dimensions (channels, number of kernels, and input resolution) using scaling factors. It also uses depth-wise-separable convolutions, inverted residual connections, and SE modules. With a range of 390 Million MACs and 5.3 million parameters (EfficientNet-B0) up to 24 billion MACs and 66 million parameters (EfficientNet-B7), EfficientNet fits into the High-Complexity Models category due to its advanced topology and high number of MACs and parameters for EfficientNet B2 to B7.

Optimized for embedded vision systems, EfficientNet-Lite (2020) [91] and MobileNetV3-minimalistic (2019) [85] simplify EfficientNet [59] and MobileNetV3 [82], respectively. These adaptations reduce multiply–accumulate operations (MACs) by removing features such as Swish activation [77] and SE modules, which are incompatible with the quantization methods [91] used to simplify complexity. Instead, they use the ReLU6 activation function [92] to maintain efficiency under quantization constraints. Both EfficientNet-Lite and MobileNetV3-minimalistic are mostly classified under Optimized for Embedded Use due to their simplicity and low resource requirements. EfficientNet-Lite models, for instance, include EfficientNet-Lite0 with 285 million MACs and 5.3 million parameters, and EfficientNet-Lite4 with 784 million MACs and 5.9 million parameters. MobileNetV3-minimalistic models, such as MobileNetV3-Small_minimalistic_1.0_224, have 65 million MACs and 2.0 million parameters [90], emphasizing their suitability for resource-constrained environments

MobileOne (2023) [93] introduced a unique architecture that separates its structure during training and inference, using specialized blocks for improved accuracy and efficiency during training that is simplified through a re-parameterization process for better embedded performance during inference. This separation allows MobileOne to maintain high performance during training and efficient deployment during inference, making it highly suitable for embedded applications. The different versions of MobileOne mostly belong to the Optimized for Embedded Use category. For instance, MobileOne-S1 has 285 million MACs and 5.3 million parameters, while MobileOne-S4 has 360 billion MACs and 14.8 million parameters.

The integration of CNNs into embedded devices has evolved to find multiple balances between model complexity and accuracy. These innovations are driven by the need to reduce computing and memory requirements while maintaining acceptable accuracy levels. This evolution has resulted in a generation of CNNs optimized for embedded systems, ranging from highly complex models requiring significant modifications for efficient integration, to models specifically designed for embedded use, emphasizing simplicity and lightness.

Optimizing the performance of CNNs in embedded systems is not limited to architectural improvements to the model. It requires an in-depth understanding of how these networks interact with the underlying hardware and software infrastructure. System flexibility, programmability, software stacks, and types of processing units are key factors in the efficiency, resource utilization, and scalability of CNN deployments. Evaluating these components enables us to better understand processing speed and energy consumption and to adapt CNN models to meet the demands of specific applications.

### 2.7. AI Processing Unit

The adaptability, efficiency, and effectiveness of CNNs are influenced by several critical aspects: flexibility, programmability, the software stack used and the types of AI processing units deployed. Each of these aspects contributes uniquely to the overall performance and applicability of CNN models in various scenarios.

#### 2.7.1. Flexibility

The flexibility of a system is measured by its ability to adapt to different computational requirements, memory needs, CNN model types, and CNN layer types. For example, the depth-wise (DW) and point-wise (PW) layers of MobileNet architectures have specific requirements [94]: DW layers are more memory-intensive and PW layers are more processing-intensive. In addition to the type of layer, the depth of a network influences the requirements for computation, memory access, and storage capacity, as the number of convolution kernels and channels and the size of the input feature maps change with the depth of the network. We have emphasized on three levels of flexibility:High-flexibility approach: This approach imposes few or no restrictions on the choice of CNNs, allowing for a wide range of models and therefore facilitating a large number of applications.Moderate-flexibility approach: This approach focuses on a specific subset of convolutional neural networks, resulting in restrictions on the models and layers available, thus reducing the scope of potential applications.Low-flexibility approach: In this approach, the focus is on a particular CNN model, which severely limits adaptability in terms of model architecture and the range of applicable use cases.

#### 2.7.2. Programmability

The programmability of embedded systems is crucial for efficiently deploying CNN models. Various complex mathematical operations are required to manage different types of layers within these models. For instance, layers such as SE use sigmoid functions [74] to recalibrate feature channels, whereas others may require soft-max functions to convert output scores into classification probabilities. However, implementing these operations on embedded vision systems can be challenging. Without adequate programmability support, developers would need to manually code these functions, as they might not be directly supported by the software, compiler, or hardware. This manual coding process introduces significant overhead, impacting both the development time and system performance. Instead of leveraging the optimized operations provided by the software stack, developers would be forced to implement custom solutions, resulting in slower execution times and reduced frame rates.

#### 2.7.3. Software Stacks

The software stack plays an essential role in simplifying the implementation of CNN models. Without these software resources, developers would face significant challenges in designing and deploying CNN models on specific hardware platforms. These tools simplify the development process by providing essential frameworks, libraries, and tools for tasks such as model training and optimization. Without them, developers would need to create these functionalities from scratch, demanding considerable time, effort, and expertise in both machine learning algorithms and hardware integration. Various software resources, such as TensorFlow [95], PyTorch [96], and N2D2 [97] offer tools for the design and occasional deployment of CNN models on specific hardware platforms. These resources simplify the development process and improve compatibility with various hardware targets.

#### 2.7.4. AI Processing Unit Types

We categorize processing units for CNNs into three levels based on their flexibility, from generic to specialized configurations (as illustrated in Figure 2).

General-purpose systems: Using architectures such as CPUs, GPUs, and FPGAs, these systems are characterized by their larger size and ability to deliver high performance across a wide range of AI processing tasks. Although they offer great flexibility and substantial computing power, this comes at the price of their increased power consumption and larger size. They are well suited to the use of full-featured software [95,96] for CNN implementation.Lightweight Systems: These systems employ specialized computing architectures like less powerful CPUs [98] or advanced NPUs [36] that focus on energy efficiency for particular tasks. Designed to achieve a balance between performance and power consumption, these architectures are suitable for environments where the highest levels of computational power are not necessary. They often use highly optimized dataflow and specialized software stacks such as TensorFlow Lite [99] and Spinnaker SDK Teledyne [100], tailored for specific, low-power AI applications.Ultra-lightweight systems: These systems employ specific computing and processing architectures that are generally designed for a single task, allowing a maximum compromise in favor of systems power consumption and at the expense of system flexibility [30].

Processing units for CNNs in embedded vision systems significantly vary in flexibility, programmability, and efficiency. By understanding the range of available options—from general-purpose systems to ultra-lightweight architectures—developers can better classify and select the most suitable solutions for specific applications, ensuring optimal performance and resource use across diverse scenarios. However, even with optimized architectures and well-matched hardware–software integration, many embedded systems face limitations due to resource constraints and specific application requirements. To address these challenges, quantization and pruning methods can be applied to further reduce the complexity of CNN models. These techniques minimize computational load and memory usage while maintaining efficient performance across diverse embedded applications.

### 2.8. Complexity-Reduction Techniques

Complexity-reduction techniques, such as quantization [101,102,103] or pruning [104,105,106,107], enable the co-design and optimization of CNNs while taking system capabilities into account. These methods reduce the computational requirements and memory footprint of models while improving hardware performance but may tolerate an accuracy drop of these models.

#### 2.8.1. Quantization

Quantization adjusts data representation dynamics, facilitating an algorithm–hardware co-design suited to constrained architectures (illustrated in Figure 5). This approach results in smaller operators and allows for more efficient use of memory as data size is decreased. Quantization can transform FP32 values into integer representations and can further reduce them to binary representations [108,109]. The transition to lower precision requires the careful selection of quantization schemes [110] (uniform or non-uniform) and parameters to balance minimizing accuracy loss with maximizing computational efficiency and memory savings.

Uniform quantization, by converting floating-point values to integers at constant intervals, simplifies hardware implementations but may not suit all data distributions.

Non-uniform quantization, with variable intervals, adapts better to specific data distributions and can more precisely manage critical model weights and activations.

Post-Training Quantization [112,113,114,115] and Quantization-Aware Training [116] are two common approaches. Post-Training Quantization applies to already trained models and offers a rapid, straightforward solution, though it may lead to a significant accuracy drop. Quantization-Aware Training integrates quantization within the training process, enabling the model to adapt its weights and structure to minimize the precision loss of quantization, often yielding better performance in precision-sensitive applications.

#### 2.8.2. Pruning

Pruning is a model-compression technique that leverages sparsity to reduce computational complexity. It does so by pruning weights, activations, or both at the same time (as illustrated in Figure 6). Sparsity can be either structured or unstructured. Structured pruning refers to the pruning of entire filters or layers, while unstructured pruning removes individual weights or activations. While pruning is an interesting attribute, taking advantage of it can be challenging when the induced sparsity is unstructured. In such cases, hardware implementations may lead to non-contiguous memory access, making unstructured pruning difficult to convert into a real acceleration. To overcome this challenge, some approaches use data-reorganization methods, such as compressed sparse column (CSC) [117], to efficiently manage and access sparse information. Iterative pruning, through its gradual refinement, allows for a more controlled reduction in model size and complexity, facilitating the identification and removal of redundancy without significantly impacting the model’s predictive capability.

Complexity-reduction techniques such as quantization and pruning are key to optimizing CNNs for embedded AI vision systems. Quantization reduces memory and computation requirements by adjusting data accuracy, while pruning improves efficiency by removing components deemed dispensable. Together, these techniques align CNNs with specific hardware capabilities, guaranteeing efficient operation within the limits of available resources.

With a complete understanding of machine vision system components—from image capture to processing optimization, memory, and processing units—we now explore the evaluation metrics that measure overall system efficiency. These metrics provide a detailed assessment of the entire inference pipeline, from initial image acquisition to final decision-making, offering insight into AI vision system performance.

#### 2.8.3. AI Vision System Metrics

Analyzing the metrics of computer vision AI systems is essential for understanding the performance characteristics and limitations of existing architectures. These metrics reveal how a system balances the trade-offs between processing speed, energy efficiency and resource utilization. The frame per second, system size, power consumption, energy efficiency, and utilization are key parameters for assessing the suitability of different architectures for different applications. By examining these factors, we can identify potential bottlenecks and areas for improvement, enabling us to guide future designs, choose the systems best suited to our challenges, or optimize them to better meet the requirements of real-world applications.

#### 2.8.4. Frames per Second

The number of frames per second (FPSs) measures the vision system’s ability to process images in one second, indicating the end-to-end processing latency from the capture of the image by the sensor to the output of the result. This measurement is essential for applications requiring real-time processing and responsiveness.

#### 2.8.5. System Size

The size of a system, often expressed in terms of the physical dimensions of its components, indicates the space occupied by the integrated circuit. A smaller, more compact system makes more efficient use of space, which is beneficial for applications where portability or miniaturization of the device is essential. In contrast, larger systems can have the advantage of incorporating bigger and more versatile components, which can improve performance but at the cost of a larger physical footprint.

#### 2.8.6. Energy Consumption (Watts)

The energy consumption of an integrated circuit is measured in watts and varies based on the performance and design of the hardware components [7,8]. Systems designed for high-performance tasks often require more energy, which can influence the choice of components and the overall system design. Conversely, circuits with a lower energy profile benefit from extended autonomy and reduced operational costs, though this may come at the expense of reduced computational power.

#### 2.8.7. Energy Efficiency (TOPS/Watt)

Energy efficiency is defined by how effectively a system performs tasks while minimizing energy consumption. Enhancements in energy efficiency can be achieved through several strategies, such as reducing the resolution of image processing to decrease data processing demands (as shown in Section 2.1), optimizing processing algorithms to lower computational complexity (as detailed in Section 2.8), or adjusting the processor’s clock frequency. The most significant improvements often result from optimizing dataflow within the CNN to maximize data reuse, thereby reducing unnecessary energy expenditure [7].

#### 2.8.8. Utilization Rate

The utilization rate of a CNN’s processing accelerator refers to the average number of computational resources actively engaged in processing tasks relative to the total available resources. This metric provides insight into how efficiently the accelerator is being used to perform computations necessary for CNN inference. However, this detailed metric is often omitted from descriptions due to the challenge of providing precise measurements.

Understanding AI vision system metrics enables a comprehensive evaluation of their performance in real-life scenarios. An analysis of frames per second reveals the system’s latency and suitability for time-sensitive tasks, while the study of system size and power consumption highlights how efficiently the architecture uses physical space and energy resources. Energy efficiency and utilization rates provide insight into how optimally the system manages its computational tasks, identifying inefficiencies that could be corrected in future iterations. By using these metrics, we can highlight the strengths and weaknesses of the architecture, enabling strategies to be developed to improve system performance, reduce energy consumption, and improve the overall design of AI-driven vision systems.

### 2.9. Concluding Insights on Metrics and Characteristics of Embedded AI Vision Systems

The successful exploitation of CNNs in embedded applications relies on an efficient balance between computational requirements, memory resources, and data bandwidth to minimize latency and reduce power consumption. Such a balance not only requires systems to be highly programmable and flexible for different CNN models and layers but also highlights the need for pre-processing and complexity-reduction techniques. These strategies are essential to enable the deployment of robust CNNs and optimize the power consumption and efficiency of the embedded AI vision system. Furthermore, the complexity of integrating CNNs into vision systems highlights a nuanced design space, in which hardware and software optimizations play a key role in meeting the diverse requirements of practical applications.

The following sections examine how near-sensor and in-sensor systems are particularly suited to these challenges, demonstrating innovative adaptations to the constraints and opportunities offered by embedded AI vision systems.

## 3. Near-Sensor AI Vision Systems

In this section, we investigate how embedded AI vision systems based on near-sensor processing adapt to the specific constraints of embedded applications. We categorize these systems into three sub-categories, as shown in Figure 7, each designed to meet distinct operational requirements.

General-purpose AI vision systems are designed for high flexibility approaches, requiring significant computing power of general-purpose processing units to process high-complexity models.Lightweight AI vision systems are designed for a moderate-flexibility approach to simpler tasks, using less sophisticated adaptable integration models and reduced computing resources with lightweight processing units.Ultra-lightweight AI vision systems are designed for low-flexibility approaches and minimal power consumption and size with ultra-lightweight processing units, using basic CNN models optimized for embedded use and minimal processing capabilities.

This section illustrates the full range of considerations, from sensor capture technologies and pre-processing algorithms to the types of neural network accelerators used. We will discuss the trade-offs involved and the software options available and finally highlight performance metrics such as power consumption and latency.

### 3.1. General-Purpose AI Vision Systems

General-purpose AI vision systems are optimized to perform a wide range of highly complex CNNs without the need for complexity-reduction techniques and with high accuracy. These systems are used in areas such as surveillance and industrial and general image processing [29,34,38]. These systems generally consume between 10 W to 30 W and offer a wide range of processing options. They enable a wide range of processing to be integrated but remain more limited than systems based on cloud computing. In certain contexts, these systems can be considered as low-energy-consumption systems, for example, when compared with the energy budget of a car.

These systems incorporate cameras with standard capture resolutions suitable for machine vision applications, ranging from 1.2 megapixels (1920 × 1080) [34] to 8 megapixels (3840 × 2160) [29]. By limiting the resolution, they reduce the amount of pre-processing required to match the capture resolution to the processing resolution, which reduces energy consumption.

They also incorporate high-performance processing units for advanced computing, such as GPUs [66,118] and CPUs [98,119].

The choice of processing units for CNN inference depends on a balance between size, FPS, and power consumption. For example, a Jetson AGX Xavier (as shown in Table 1) offers 32 TOPS of AI performance. It might therefore be able to deliver more frames per second than processors such as the Khadas VIM4 (3.2 TOPS), but with much higher power consumption.

GPUs are based on highly programmable parallel architectures, originally designed for graphic rendering. However, they have proved highly efficient for processing the mathematical operations of neural networks because of their ability to perform many operations in parallel. Processors, in contrast, are designed to perform sequential, general-purpose tasks.

In GPU-based systems [66,118], the GPU handles the bulk of parallel computing while a CPU orchestrates these tasks, ensuring efficient operation. This combination exploits the strengths of both components, delivering high performance for demanding applications. Furthermore, the presence of a CPU makes it easy to implement pre-processing techniques, improving overall system efficiency. Meanwhile, CPU-only systems [98,119] offer lower power consumption but at the cost of reduced performance for parallel tasks. However, they still benefit from the flexibility of CPUs, enabling various pre-processing techniques to be used. Despite these differences, GPU- and CPU-based systems retain a high degree of flexibility, capable of efficiently handling a wide range of computational tasks. Combined with multi-gigabyte memories, both types of systems are capable of handling all CNNs presented in Figure 4 without the need for complexity-reduction techniques.

GPU-based and CPU-based systems can operate at a full dynamic range (FP32) to preserve accuracy while having the option of adopting a lower dynamic range to further improve hardware performance. The processing units benefit from full operating systems (such as Ubuntu in [29,34,38]) that allow them to leverage advanced software libraries, such as Tensorflow Lite [99], OpenCL [26], CUDA [27] and TensorRT [28].

By automating the implementation of CNNs, users can select their application. These platforms facilitate the selection and deployment of CNNs, allowing the use of pre-trained models with or without complexity reduction, as well as the rapid implementation of custom networks.

General-purpose AI vision systems are flexible systems with few constraints in terms of size and power consumption. Some systems have more stringent requirements in terms of size and power consumption, to be integrated into more demanding environments. To meet these needs, lighter, more specialized and optimized systems can be used. In the next subsection, we will examine in detail the consequences of reducing the overall size of the AI-integrated vision system in terms of performance and flexibility.

### 3.2. Lightweight AI Vision Systems

Lightweight AI vision systems are optimized for flexible embedded IoT applications, requiring low energy consumption and small in size [35,36,39,40,41,42,43,44]. Achieving this often involves trade-offs to reduce the system size to less than 100 cubic centimeters, encompassing capture, pre-processing, and AI processing.

These trade-offs impact overall flexibility and performance. Therefore, these systems use AI hardware that is specific to a CNN class called compact or lightweight CNN [24]. This class of CNN model prioritizes energy efficiency while maintaining accuracy (as shown in Table 2). Instead of relying on more flexible and programmable GPUs, they use NPUs and TPUs or small CPUs.

These systems encompass a broad spectrum of camera options. For instance, the OAK-1 [36] offers cameras up to 12 MPs at 30 FPS, and smaller systems like JeVois [43] offer a 1.5 MP at 60 FPS camera. These variances may be attributed to different application needs.

JeVois [43] provides a flexible and highly integrated system founded on a small, versatile processor Allwinner A33 [121], while the OAK-1 [36], leveraging a Myriad X [122], efficiently processes CNNs, incorporating pre-processing directly within the Myriad X [122]. In both systems, the cameras can capture both color and grayscale images, further simplifying CNN pre-processing.

Others, such as [123]’s system, suggest integrating pre-processing directly into the NPU, reusing the hardware already present in the NPU to create a more compact and integrated system.

Lightweight AI vision systems are strategically engineered to balance the integration of CNN processing with compact size. These systems range from the more powerful, such as AIVision [41] with 2 GB of memory, to constrained systems with as little as 24 MB [35]. Those with minimal memory are specifically tailored for CNN models that exploit complexity-reduction techniques. They engage in a pronounced trade-off between system flexibility and lower FPS performance levels to optimize resource utilization.

An example of a constrained system is Firefly Teledyne [35]. Equipped with a 1 TOPS NPU, it can achieve 12 fps for Mobilenet V1 1.0 224 [58] and 4 FPS for Inception v1 [124], as detailed in the Teledyne FireflyDL documentation [125].

In the context of constrained systems, the limits of CNN model embeddability are reached, even with complexity-reduction techniques. The JeVois [43] system illustrates this, where the model itself is modified for integration, managing 7.6 FPS using a carefully optimized MobileNet v1 224 0.5 [58] with only 12 of its 18 layers, as demonstrated in [126].

Each lightweight AI vision system uses a constrained software stack tailored to its hardware constraints, providing details on the CNN models it can support. For example, the AIY Vision Kit [41] supports TensorFlow Lite models like MobileNetV1 (input size 160 × 160, depth multiplier 0.5) and MobileNetV1 + SSD (input size 256 × 256, depth multiplier 0.125). Similarly, JeVois [43] uses an optimized stack for models like MobileNetv1_0.5, allowing reduced layers for efficiency. The Firefly DL [35] with Spinnaker SDK supports MobileNet V1 1.0 224, achieving 12 FPS at low power. These stacks highlight the trade-offs between performance and resource constraints inherent to each platform.

Some applications require even more stringent constraints, requiring highly compact embedded systems with power consumption ranging from under a watt down to mere nanowatts. In these contexts, commercialized NPUs, TPUs, and generic CPUs/GPUs are not suitable due to their size and power consumption.

### 3.3. Ultra-Lightweight AI Vision Systems

Therefore, we need to consider dedicated and optimized architectures, which are not commonly found in commercial products because the work is carried out by university researchers or targets a specific task. These systems can be found in ultra-embedded drones, ultra-low-power surveillance, and ultra-low-power IoT.

Ultra-light embedded AI vision systems (as indicated in Table 3) meet the need for ultra-low-power in size-limited applications.

For example, these systems are used in drones for real-time navigation and decision-making [33], image recognition [45], always-on facial recognition [32,46], and object recognition [47].

In contrast to the systems described Section 3.1 and Section 3.2, which tend to rely more on COTS components, ultra-light embedded AI vision systems often include a highly specialized part of the system, designed either for a specific task or to minimize size and optimize power consumption.

Ref. [33] uses the PULP GAP8 architecture [30], an RISC-V multi-core processor designed for ultra-low-power embedded applications with a focus on efficient execution of CNNs. Since the GAP8 consists of RISC-V cores, it is a programmable architecture that can adapt to new developments in CNNs. However, as a trade-off for these optimizations, the architecture has limited flexibility. The benchmarks for this circuit show that it supports a limited set of layers with small convolution sizes [127] (such as convolutions 5 × 5, stride 2, and the max pooling layer). Larger convolutions are not necessarily required as most CNNs use 3 × 3 or 1 × 1 convolution [86]. Techniques such as convolution factorization and depth-separable convolutions enable the use of deep CNN with small convolution sizes, which can still achieve efficient and powerful performance in ultra-embedded applications. The architecture works for 16b fixed point data, which are quantified data, enabling it to reduce the complexity of the calculation.

In [32,46], the processing architecture, called the CNN processor (CNNP), is highly specialized for research purposes. It features a 4 × 4 processing element (PE) Array with distributed memory for CNN weights and a 16-way MAC Array for processing 16 parallel 16-bit data elements. The PE design incorporates max, sum, pooling, or shift operations into its ALUs. This highly optimized architecture enables near-sensor CNN inference for simple CNN layers (convolution and fully connected). The implementation relies on the use of linearly separable filters and convolution factorization to reduce the complexity of CNN layers.

The dedicated neural architecture (NE) in [45] is specifically designed to accelerate CNN operations and enable hierarchical image recognition. Among the main components of this NE architecture is the NE Control Executor (NCX). NCX is an NE-specific RISC processor that drives the processing element by executing instructions from the NE instruction memory. It communicates with other IP blocks through high-performance BUS and interrupts them. The PE is the computational core of the NE, comprising an eight-bit multiply–accumulate (MAC) array and buffers. Buffers for weights, biases, and inputs/outputs ensure sufficient bandwidth for optimal use of the MAC array.

These various approaches to integrating CNN processing closer to the sensor are accompanied by pre-processing methods to reduce data complexity. For instance, [32,46] use Analog–Digital HAAR-Like Face Detector, which is an ultra-low-power hardware specifically designed for face detection using a Viola–Jones filter-based face-detection algorithm [61]. Ref. [33] does not use pre-processing and shares the cache memory of the captured data with the inference circuit, allowing for maximum optimization in terms of both computing time and system size by eliminating the need for pre-processing circuitry. [45] uses offline compression methods and a Weight Decompressor that decodes compressed weights from NE’s shared memory and loads them into the weight buffer on the fly.

To minimize the need for pre-processing or even eliminate it altogether, image sensors used in these systems have been reduced in size and capture resolution, sometimes even matching the processing resolution of the CNN. These grayscale image sensors come in various formats, ranging from VGA for [45] with 32 × 32 processing down to QQVGA for [47]. While specific information on the image sensors used is limited, [33] integrates a HiMax CMOS image sensor [128]. This ultra-low-power sensor has a QVGA resolution (60 FPS at 4.5 mW) and is suitable for a 200 × 200 processing resolution. The HM01B0 model, which has a small size of 2.5 mm × 2.5 mm, is likely to be the chosen sensor, highlighting the strong emphasis on ultra-low-power and small size in these systems.

These systems rely on low-capacity memories, ranging from a total of 192 MB for [33] to 1.3 MB for [32,46] down to 3840 bits for [47]. Such small memory sizes allow for a reduction in the memory’s surface impact. However, it may reduce the circuit’s autonomy in cases where the weights must be stored in a host capable of integrating the vision system. In terms of computing performance, these systems operate in an order of magnitude lower (GOPS instead of TOPS) than other near-sensor processing categories due to the extreme embedded constraints.

In terms of performance, [32,46] implement their facial recognition system at 1 FPS for 0.62 mW using a custom network with four convolutional layers and a fully connected layer. Ref. [33] proposes a DroNet [129] CNN implementation, which is a Resnet8 variant, quantized and optimized for performance, achieving 6 FPS at 64 mW.

### 3.4. Conclusion on Near-Sensor Processing AI Vision Systems

Near-sensor AI offers a wide range of solutions for the diverse needs of AI vision. We examined three subcategories of near-sensor machine vision systems, each addressing varying levels of application requirements. General-purpose AI vision systems are designed for complex applications requiring significant computational power, flexibility, and programmability to process advanced CNN models with minimal embedded constraints. Lightweight AI vision systems are adapted for flexible embedded applications that are manageable with a limited model range and reduced computational power to address stronger constraints. Finally, ultra-lightweight AI vision systems are intended for ultra-embedded applications needing ultra-low-power consumption and highly compact size, featuring ultra-lightweight CNN models and minimal computational power to satisfy embeddability constraints.

All these vision system models rely on the interconnection of separate subsystems, whether they are COTS or more specialized systems. Due to the interconnection of separate subsystems, the cost and time of transferring data between the sensor and the processing unit, or between the processing unit and the memory, reduces energy efficiency and increases latency. To overcome these challenges, we will explore in the next section how in-sensor processing AI vision systems offer promising solutions.

## 4. In-Sensor AI Vision Systems

Embedded AI vision systems combine many complex elements, including cameras, pre-processing systems, and CNN inference systems such as CPU/GPUs, NPUs, and TPUs. However, the physical distance between image capture and CNN processing presents a real challenge for performance improvements. This not only introduces cost and latency but also affects energy efficiency. To overcome near-sensor challenges, intelligent sensors with in-sensor processing capabilities have been developed with more parallel processing of information within the sensor itself. This challenge has also been addressed in the field of general-purpose in-sensor processing [53,54,55], leading to insights into designing vision systems that integrate AI directly into the sensor.

We have highlighted two main types of in-sensor processing AI systems (as shown in Figure 8): optimized NPU-based processing (ONP) and highly parallel distributed processing (HPDP). These systems demonstrate intriguing trade-offs between data access parallelism, energy efficiency, and latency, presenting a valuable alternative for applications requiring low latency and high energy efficiency.

This section is organized as follows: it begins with an introduction to 3D integration technologies, fundamental to both ONP and some HPDP systems; it continues with a detailed examination of ONP systems; it then investigates 2D HPDP systems utilizing CNN processing in the focal plane; finally, it delves into HPDP systems that leverage 3D integration technologies, illustrating how these enable highly parallel CNN inference while mitigating some 2D HPDP system limitations.

### 4.1. Introduction to 3D IC Vision Systems

Three-dimensional integration technologies [130] are circuit-manufacturing technologies that enable several circuits of CMOS transistors to be stacked on a single circuit. Several circuits are stacked instead of being placed side by side, which increases the density of components on the same circuit and the processing capacity for the size of the circuit. In a 3D CMOS stack, the layers are joined together by vertical connections called Through Silicon Via (TSV) [131] or Cu-Cu wafer bonding connections [132] that allow data to flow between the layers. The choice between TSVs and Cu-Cu wafer bonding connections depends on design specifications, required performance, cost, and manufacturing constraints. When stacking more than two layers of ICs, a combination of TSVs and Cu-Cu connections or TSVs only can be used, depending on the performance requirements, connection density needs, cost, and manufacturing constraints [64]. In the context of complex systems such as vision systems, stacking layers beyond three layers [64] is a technology that is still being developed [133]. Three-dimensional integration technologies offer the ability to specialize each CMOS layer into a specific technology node, with a different lithography size for each layer, allowing them to be tailored to the desired processing requirements (as shown in [134]). However, using 3D integration technology to manufacture CMOS circuits involves high production costs and increased complexity compared with the manufacture of conventional 2D circuits.

Three-dimensional integration technologies in image processing make it possible to design architectures in which a layer dedicated solely to the photosensitive area of the pixels is superposed to a processing layer. This configuration not only improves pixel density in the camera but also considerably reduces the latency time seen in near-sensor systems between capture and CNN processing. By tightly integrating these two layers, the time taken for data to travel from the sensor to the processing unit is minimized, improving overall system efficiency (FPS and power consumption) [64,134,135,136].

In the rest of this section, we will see different types of architectures, their architectural specificities, CNN inference process, and opportunities for improvements.

### 4.2. Optimized NPU-Based Processing

A specific ONP architecture [31] incorporates an NPU in a dedicated CMOS layer beneath the pixels (as shown in Figure 8 and Table 4). This design enables CNNs to be executed directly in the sensor, meeting integration requirements while balancing processing efficiency, frame rate, and power consumption. The system is composed of two specialized layers manufactured using different CMOS lithography technologies: the upper 65 nm layer for the pixels and the more advanced lower 22 nm layer for processing. This distinction makes it possible to optimize the performance of each layer according to its specific function. Using 22 nm technology for processing can improve density and energy efficiency, and selecting 65 nm for pixels may be a deliberate choice for other considerations, such as pixel sensitivity and cost.

#### 4.2.1. CNN Inference Process

Ref. [31] presents a vision system dedicated to CNN processing. This system has strong similarities with the near-sensor AI vision systems described in Section 3, since it combines the functional components of near-sensor systems within a 3D stacked solution. The system includes a camera for image capture, offering a resolution of 12MP for 1.5 μm × 1.5 μm industrial pixels, which is between the performance of the general-purpose AI vision systems and lightweight AI vision systems.

Once the image is captured, it is sent to an ISP in the second CMOS layer. This architecture, unlike near-sensor systems, enhances parallelism between pixels and the ISP by adding a column of Analog-to-Digital Converters (ADCs) that convert the analog signals from the pixels into digital data at the foot of the pixel matrix. This not only increases the data bandwidth between pixels and ISP (as described in [64]) but also maintains high parallelism with a compromise on the area designated for the ADCs.

Similar to certain near-sensor systems, an ISP is used to pre-process the captured image before inference. The capture is conducted in two possible modes: 12.3 MPs for 30 FPS processing and 3.1 MPs for 120 FPS. The CNNs processed in this AI vision system use an input image size of 224 × 224. The system accordingly carries out unspecified transformations on the image prior to proceeding with inference.

The system’s memory is constrained to only 9 MB, which necessitates the use of 8-bit quantization. Despite this constraint, the quantized approach is sufficient to store all the weights of a CNN, such as Mobilenet_V1_1.0_224_quant [137], similar to those in lightweight systems like [35]. While no specific software stack has been delineated, the obtained results indicate moderate flexibility; the system is equipped to execute inference using several cutting-edge quantized CNN models, including Mobilenet_V1 [58], MobileNet_V2 [86], and Inception_v1 [124]

Once pre-processed, the image has two pathways within the system. It can be transmitted directly outside the device via a MIPI interface or processed further by an NPU. In this particular system, the NPU functions as a DSP with two specialized computing cores. Based on the system’s design, one can surmise that the first core, focusing on data reuse and intensive computing, may be suitable for handling conventional convolution layers or point-wise layers within networks featuring depth-wise separable layers. Similarly, the second core, tailored for intensive memory accesses, could be an apt choice for depth-wise layers or fully connected layers. This strategic specialization in accelerator design can also be found in lightweight systems, as highlighted in [24].

The system described in [31] achieves TOP1 accuracy of 70% with Mobilenet_v1, comparable to GPU-based inference. Offering two processing modes with frame rates of 30 or 120 frames per second, it consumes 278.7 mW and 379.1 mW in the 12.3 MPs and 3.1 MP acquisition modes, respectively.

This performance is based on 3D integration technology, which reduces latency and optimizes power consumption. Although the system structure is similar to that of near-sensor (capture, ISP pre-processing, and CNN processing with an NPU), 3D integration adds efficiency. By adapting the technology nodes and improving the parallelism between capture and CNN processing, the system’s power consumption performance matches that of ultra-light systems, offers the flexibility of light systems, and far exceeds the FPS performance of general-purpose systems.

Overall, [31] illustrates how 3D integration can combine high performance, low power consumption, and flexibility in CNN model selection, a breakthrough that is echoed in other works such as [138], which discusses the load balancing of a complete DNN (head and backbone) in a combined in-sensor 3D stack and near-sensor system. This advance highlights the potential of 3D stacking in creating highly integrated and efficient systems for CNN inference.

#### 4.2.2. Opportunities for Improvement

Several aspects of the system present opportunities for improvement. The current circuit size of 7.558 mm × 8.206 mm could be reduced by eliminating the unused portion of the photosensitive matrix, which is currently necessary to match the dimensions of the underlying layer in a 3D assembly. By integrating components such as the ISP within the NPU, as suggested in [25,123], the overall size of the circuit could be minimized. This approach would allow the total circuit size to be determined by the largest of the fused layers, rather than maintaining uniform dimensions across all layers. This integration allows for a trade-off between processing latency and component fusion, leading to a reduction in the surface area needed for memory, ISP, and accelerator components.

Access to the photosensitive matrix also poses a challenge. While [31] provides greater bandwidth than traditional near-sensor systems, the data transfer from the photosensitive array to the ISP remains constrained. Other systems not specifically designed for CNNs show how a highly parallel processing architecture could further optimize these accesses. These research-led innovations and the possibilities they offer for new developments in computer vision systems will be examined in the next subsection.

### 4.3. Two-dimensional In-Pixel Processing

Two-dimensional vision systems with processing in the focal plane, also known as Pixel Processor Array (as shown in Figure 8 and Table 5), are a set of sensors that integrate mixed analog/digital processing within each pixel or smart pixel. These systems have been designed for performing inherently spatial and parallel image processing tasks such as HDR rendering [60], HAAR filters [32], and edge detection [48,49].

These vision systems are designed with a balance between the size of each pixel’s photosensitive area and the integration of analog/digital processing within the pixel. Comprising compact pixel arrays, such as 256 × 256 configurations, every pixel is equipped with the necessary components to perform basic computations, including an ADC, binary and analog registers, a small amount of memory, and elementary operators for addition and shifting. The pixel size of these systems is much larger than that of other vision systems because they include capture and processing in a single structure called a pixel processor. For example, the pixel processor size in [53] is 32 μm × 32 μm in 180 nm CMOS technology. These vision systems consist of a grid of pixel processors. They can operate independently or cooperatively and can also perform systolic calculations. Numerous studies [50,51,139,140] have demonstrated the feasibility of incorporating CNNs into 2D vision systems with focal plane processing. These studies provide digit classification applications using one- to three-layer CNN models on the MNIST [141] database.

To understand the advantages and limitations of these systems, we will detail their inference process.

#### 4.3.1. CNN Inference Process

During the image capture, [51] focuses on the precise positioning of the image on the imager, while [139] requires optimal alignment for inference. After capture, the image is thresholded to minimize the dynamic range of the data and then stored in the imager. As these implementations are proof-of-concept demonstrations of the integration of CNNs into sensors, the presence of these limitations in capturing is to be expected, reflecting the innovative and complex nature of the approach.

Once the image has been captured, the implementation of the CNN processing is similar to all the examples described in the literature. The objective is to physically place the CNN weights in the right place on the array to perform the calculations required for each CNN layer. In order to run multiple convolution kernels in parallel, the input feature map is replicated [50,51,139]. The replicas are spread over the imager’s surface to allow simultaneous inference of multiple convolution kernels. As a result, a set of outputs from a CNN layer are generated in parallel. After convolution kernel applications, it may be necessary to perform accumulations, particularly when there are multiple feature maps at the input of a CNN layer. The data are then shifted and summed over the entire Pixel Processor Array [139].

Both [50,139] use 28 × 28-pixel images to infer a LeNet [73] type network using the SCAMP-5 architecture [53]. Specifically, [50] uses this architecture for two convolution layers and one fully connected layer executed off-chip. In contrast, [139] applies three layers, including two convolution layers and one fully connected layer, directly within the SCAMP-5 architecture. The CNNs used are drastically quantized to a binary [50,140] or ternary [37,139] data dynamic range. This allows for saving memory and computational resources by replacing convolution operations with equivalent addition and shift operations, equivalent to division by two. These CNNs apply techniques to reduce complexity without significant accuracy loss for the given task [108]. Another optimization involves performing part of the calculations in the analog domain [50,139,140]. This approach, which introduces noise into the data, is corrected in the training of the CNN as shown in [50].

Ref. [139] achieves a high-performance result of 210 FPS, with an architecture comprising three layers: CONV1: 16 filters of size 4 × 4; CONV2: 16 filters of size 4 × 4; and FC1: 256 weights × 10. This approach consumes approximately 2 watts for full inference. In contrast, [50] reports even higher performance, reaching a speed of 2260 FPS with a simpler model consisting of a single layer with three 3 × 3 convolutions.

#### 4.3.2. Opportunities for Improvement

Scaling these architectures is complex because the amount of memory and computing elements is limited by the surface area of the sensor, or even by the surface area of the pixel when these elements are integrated into each pixel [53].

The parallelism of the calculation is limited by the size of the matrix and the size of the input feature map. When the size of the input feature maps from the CNN does not match that of the sensor, it is necessary to partition the CNN [139], store the results awaiting processing, and orchestrate the data movements, all of which require significant memory requirements and data movements. In addition, moving from one layer of the model to another also requires a large number of data movements (kernel and input feature map) to be organized, which can be costly in terms of latency and data movements. Another limitation concerns convergence trees, as each convolution is the application of a kernel to a set of input feature maps. After these operations, it is necessary to make an addition tree of all the applications of the kernel to the input feature maps. This spatial operation is not trivial on pixel matrices [50,51,139].

In addition, CNN weights generally occupy several megabytes of storage and are difficult to transfer to embedded solutions. Two-dimensional in-pixel processing approaches address this challenge by storing the weights in external memory separate from the circuit. As a result, full and independent integration of CNNs into these systems often requires a host system [139] capable of orchestrating the different CNN layers, turning the system into a co-processor. This dependence on a host system must be taken into account if an autonomous solution is to be developed for the future. In summary, 2D vision systems with focal plane processing offer interesting prospects for highly parallel processing of CNNs, but their implementation poses challenges in terms of scaling, layer switching, portability, and autonomy of the solution. In the next subsection, we present vision systems that use 3D stacking technologies that overcome some of the limitations of 2D vision systems with processing elements in the focal plane. This camera stacks several CMOS layers specialized in specific tasks, such as capture, analog-to-digital conversion, or data processing, while establishing efficient and highly parallel communication between them.

### 4.4. Three-Dimensional Focal-Plane Array Image Processor Chip

Refs. [62,142] introduce a programmable neighborhood processing vision system using 3D integration technology (as shown in Figure 8 and Table 6). The system is built on two layers fabricated in 130 nm 1P6M CMOS and features an array of 3D macropixels (MPXs), each responsible for a 16 × 16 group of pixels (12 μm × 12 μm each). These macropixels are unique three-dimensional pixel structures that share hardware processing resources. They allow for a balanced trade-off between parallelism of data access and processing integration within the sensor, enabling highly parallel processing. Inside each MPX, there are 16 programmable processing elements (PEs) equipped with an 8-bit ALU for tasks like multiplication, addition, shifting, etc. Memory within the system is organized across two levels: the local memory (RF) at the MPX level, fed directly by the ADCs of the pixels, and a second embedded SRAM level that can read and write to the first. The system supports two image-acquisition modes: 256 × 192 and 1080 × 768.

For the remainder of this subsection, we will discuss the implementation of CNNs made in [23] on [62,142]. Specifically, we will compare this system to the implementations described in Section 4.3, as those systems use similar CNN models.

#### 4.4.1. CNN Inference Process

Ref. [23] introduces the first implementation of a CNN inference pipeline on a 3D IC focal-plane array image processor chip, adopting a LeNet-style CNN. The network consists of two convolutional layers and two fully connected layers, specified as follows: CONV1: 16 filters of size 4 × 4, CONV2: 24 filters of size 5 × 5, FC1: 384 × 150, and FC2: 150 × 10. The system also accommodates ReLU functions and operates with a four-bit quantized CNN from [97].

The process begins by inserting a 24 × 24 image into a specific position on the MPX matrix, duplicating it to apply the CONV1 filters in parallel. For the CONV2 layer, intelligent distribution enables simultaneous calculation across 16 previous feature maps. The resulting outputs are then aggregated using an adder tree to form the final results.

The FC1 layer transforms these outputs into a feature vector of size 384 using a five-step algorithm, with 150 hidden neurons performing weighted computations. The FC2 layer follows the same methodology as the FC1 layer.

The results demonstrate the full implementation within the sensor, with the system exploiting different levels of parallelism to complete the two convolutions and two classification layers. The total classification time was 3.8 ms at a frame rate of 265 FPS. This performance compares favorably with that of [139], which reached a rate of 210 FPS for a three-layer CNN with one FC layer.

Similar to the system explored in Section 4.3, ref. [23] designs a CNN inference method distributed across a group of PEs, located directly in the sensor for enhanced parallel processing. What distinguishes [23]’s approach is its ability to perform a wider variety of operations per pixel, striking an interesting balance integration of processing inside the pixels and computational flexibility. This balance paves the way for more complex CNN models, characterized by additional layers, greater computational dynamics, and an increased number of operations within each layer.

In contrast, when considering the architectures of [50,139], the amount of memory and computational components is not limited by sensor or pixel dimensions. As these components are integrated in a separate layer beneath the photosensitive matrix, the parallelism of calculations is not subject to the constraints of the size of the photosensitive matrix. As a result, future iterations of the architecture design can be scaled effortlessly by increasing the number of MPXs and PEs.

#### 4.4.2. Opportunities for Improvement

Ref. [23] also highlight areas where the approach could be refined. While the convolutional layer implementation proves effective when weight kernels operate simultaneously on various subsets of input data, any reduction in parallelization on a given layer can lead to a decrease in efficiency.

To enhance performance, it is crucial to evaluate and propose new dataflow that can be used in a matrix of processing elements. This optimization improves the computational efficiency by increasing the utilization rate of the PEs, potentially enabling the parallel inference of multiple layers.

It is also necessary to reconsider the architecture to find new trade-offs for integrating much larger models, making [23] comparable to other architectures that use 3D integration technologies with state-of-the-art networks like [31].

Finally, [23] notes potential scalability restrictions arising from the data movements required to provide the computational elements. To overcome this challenge, the introduction of fast data-transfer mechanisms would be essential.

### 4.5. Conclusion on In-Sensor Processing AI Vision Systems

In-sensor approaches represent a more integrated and therefore more complex strategy in the evolving landscape of vision systems. These approaches, while more difficult and expensive to implement, have the potential to significantly exceed the capabilities of near-sensor systems. More specifically, in-sensor solutions such as [31] represent a first step towards greater integration. The 3D stack approach demonstrates that a design similar to the near-sensor one (including the sensor, pre-processing, and NPU) can be sufficient to achieve significant performance gains. Innovation in this area is not limited to 3D integration. The methodology of 2D vision systems may have more constraints than a 3D approach such as [23]. However these are systems for generic image processing, and their innovation lies in their ability to inspire new dataflow architectures for image processing directly in pixels. These new dataflow architectures offer the most promising prospects for outperforming conventional near-sensor and in-sensor ONP [31] approaches. This is particularly true in scenarios where data access parallelism remains limited between the capture layer and the processing layer, as well as between the NPU and memory. Collectively, these observations point the way forward, with in-sensor approaches promising to transcend existing limitations. The balance between integration complexity and performance gains is likely to continue to shape advances in this area, with innovative dataflow designs and 3D stacking techniques offering potential directions for future exploration and development.

## 5. Conclusions

The evolution of artificial intelligence vision systems is manifesting itself in a rich diversity of methods and approaches, with near-sensor and in-sensor systems representing two important paths of exploration and innovation.

### 5.1. Near-Sensor Systems

Near-sensor systems have paved the way for greater efficiency and performance by placing processing close to the sensor, reducing the need for time- and energy-consuming data movement. This design has proved effective in various applications, offering a more energy-efficient, low-latency approach to data processing. Unlike cloud-based systems, which consume more than 100 W and are hosted on large servers, near-sensor systems are more limited but highly flexible. They operate at 30 W or less and range in size from 500 cm^3^ to a few millimeters, making them ideal for embedded applications.

General-purpose machine vision systems, such as [29] equipped with NVIDIA Jetson AGX Xavier, offer significant computing power (up to 32 TOPS) while consuming between 10 W and 30 W and being approximately 500 cm^3^ in size [29,34]. These systems are ideal for highly complex tasks that require advanced CNN models.

Lightweight machine vision systems, such as those based on Myriad 2 and Myriad X processing units, strike a balance between flexibility and efficiency, consuming between 1W and 5W and fitting into sizes as small as 1.65 cm^3^ to 54.5 cm^3^ [35,36,43], are ideal for moderately complex tasks in embedded IoT applications.

Ultra-lightweight machine vision systems, such as those using PULP GAP8 architecture or specialized neural engines, operate with ultra-low-power consumption (as low as 0.17 mW) and highly compact dimensions (down to a few millimeters) [32,33,45]. These systems are designed for ultra-embedded applications where minimal power consumption and small size are essential.

By integrating processing capabilities close to the sensor, these systems improve overall efficiency and performance, making them crucial for the advancement of machine vision technologies. This approach significantly reduces power consumption and latency, offering a versatile and efficient solution for a wide range of embedded applications.

### 5.2. In-Sensor Systems

In-sensor vision systems have taken the concept of near-sensor vision systems a step further by integrating computing elements directly into the sensor. This approach, including 3D integration technologies, has transcended some limitations of near-sensor systems, offering significant advantages in terms of power consumption and latency. In addition, 2D and 3D architectures in the sensor vision system category have presented unique advantages and challenges, highlighting the sensitive balance between integration complexity, performance, and highly parallel processing.

The article investigated both paradigms, highlighting innovative techniques, success stories, and potential areas for improvement. Systems such as that described in [31], which integrate NPUs beneath the pixel array using 3D stacking, demonstrate that high performance can be achieved with ultra-low-power consumption (278.7 mW to 379.1 mW) and small size comparable to ultra-lightweight vision systems. This architecture optimizes the parallelism between pixel capture and processing, significantly reducing latency and improving overall efficiency.

In-sensor vision systems, such as the 2D focal plane processors presented in [53], take advantage of highly parallel processing within the pixel array. These systems integrate basic computational elements into each pixel, enabling the efficient execution of simple CNN models directly at the sensor level. For example, [139] demonstrates a 2D vision system achieving 210 FPS with a three-layer CNN, consuming approximately 2W.

In addition, 3D focal plane image processors, such as those described in [62], use advanced 3D stacking to integrate multiple layers of CMOS circuitry, enhancing processing capabilities and parallelism. These systems, like the one implementing a LeNet-type CNN in [23], dramatically improve performance, with FPSs up to 265 FPSs, while maintaining a compact size.

In conclusion, machine vision systems that process near-sensors and in-sensor represent a significant advance in AI vision system integration and efficiency. By minimizing data transfer and exploiting advanced 3D stacking technologies, these systems achieve unprecedented levels of performance and energy efficiency. As research and development in this field continues to evolve, we can expect even more sophisticated and high-performance machine vision systems that will push back the boundaries of what is possible in embedded applications.

## 6. Perspectives

The future of embedded AI vision systems lies in innovative architectural designs that overcome current boundaries. As computational demands increase, rethinking dataflow architectures and their integration with 2.5 and 3D stacking technologies offers new possibilities for reducing latency and power consumption. This exploration requires a nuanced understanding of the interaction between hardware configurations and algorithmic mapping. These advances will lead to breakthroughs in efficiency and performance, paving the way for cutting-edge applications in artificial intelligence.

Rethinking the design of dataflow architecture [7,8] for 3D stacking systems offers significant potential for innovative dataflow to reduce latency and power consumption. A promising approach involves co-optimizing hardware space and mapping space [143,144]. Hardware space exploration focuses on configuring the physical aspects of the system, such as the number of processing elements and memory sizing [145]. This approach allows for identifying the optimal hardware setup to efficiently meet computational demands. In contrast, mapping space exploration addresses the assignment of computations to the hardware, including loop tiling, loop order, and computational parallelism [146,147]. It seeks to optimize the execution flow to enhance overall performance. By integrating these methods, it is possible to design high-performance architectures through systematic exploration of vast design spaces, balancing constraints on latency and power consumption. Although these approaches are not specifically targeted at 3D stacking architectures, they are highly beneficial for the design exploration of CNN accelerators and significantly simplify the complex process of developing efficient and effective AI vision systems.

The evolution of 3D stacking techniques represents a promising area of innovation. Stacking up to or beyond three layers is being actively explored, particularly by the in-memory computing community [57] and major companies, such as Sony [31,64] and Alphabet [138]. These advances promise to significantly enhance conventional systems by improving data-transfer rates, reducing latency, and increasing energy efficiency. Future applications could include more efficient AI vision systems, advanced IoT devices, and high-performance computing solutions.

The use of 2.5D design technologies represents an interesting possibility for vision systems integrating AI processing, offering a balanced alternative between near-sensor and in-sensor technologies. This technology uses silicon interposers to integrate heterogeneous technologies on a common substrate, exploiting TSVs [148]. Compared with traditional 2D systems, 2.5D packaging enables higher integration density and performance thanks to a smaller interconnected pitch [149]. The use of fine-pitch interposers improves the efficiency of the interconnect network, reducing latency and increasing bandwidth, which could optimize CNN processing [150]. The interposer can be either active or passive. Active interposers offer integrated logic that further enhances data processing capabilities, while passive interposers are simpler and cost-effective, primarily serving as connection facilitators between components in a 2.5D integration. In terms of thermal management, 2.5D systems offer advantages over 3D architectures by dissipating heat more efficiently, which is crucial for computer vision systems subject to high computational loads [151]. In addition, the reduced complexity and cost effectiveness of 2.5D systems compared with 3D integration could enable efficient heterogeneous technological integration of an image sensor and CNN accelerator on the same substrate, thereby reducing power consumption and improving overall performance [152]. This approach also enables known designs to be reused, which can speed up development and reduce costs [153]. However, it is important to note that 2.5D systems, while promising, are not without their challenges. The complexity of manufacturing silicon interposers and the associated costs can represent obstacles [148]. In addition, thermal efficiency, while better than that of 3D systems, remains inferior to that of fully 2D solutions. A critical and ongoing assessment of these factors is required to maximize the benefits of this technology in machine vision applications.

The interaction between processing near-sensor applications for intensive calculations such as fully connected layers (head) and feature extraction in-sensor convolutional layers (backbone) poses a major challenge. Innovative partitioning algorithms [138,154] could provide better insight into the workload distribution, balancing approaches close to the sensor and within the sensor. This complexity lies in the need to develop an algorithm that efficiently partitions CNNs between their convolutional head and their backbone. A high data-transfer rate between processing inside and close to the sensor is unavoidable, as all feature maps must be transferred to another chip to potentially implement part of the backbone, fully connected layers, or even several fully connected layers to generate multiple feature representations or outputs for various applications [33,138]. The efficient management of this massive data transfer allows us to avoid bottlenecks and ensure minimal latency while maintaining low-power consumption.

Finally, the design of vision systems capable of inferring attentional-based models [17,18] presents another set of challenges. Despite their remarkable capabilities, these models remain voluminous. Their optimization will require a combined algorithm–architecture approach. An efficient implementation of the types of layers used, adapted to their complexity, will allow their advantages to be fully exploited while managing spatial and energy constraints.

The challenges of designing efficient computer vision systems require a comprehensive approach that balances hardware capabilities and computational requirements. By exploring new architectural paradigms and adopting emerging technologies, we can overcome current limits and push the frontiers of what artificial intelligence systems can achieve. The journey toward smarter, more efficient systems continues, driven by innovation and a commitment to meeting the complex challenges of modern AI vision system architectures. 

## Figures and Tables

**Figure 1 sensors-24-05446-f001:**
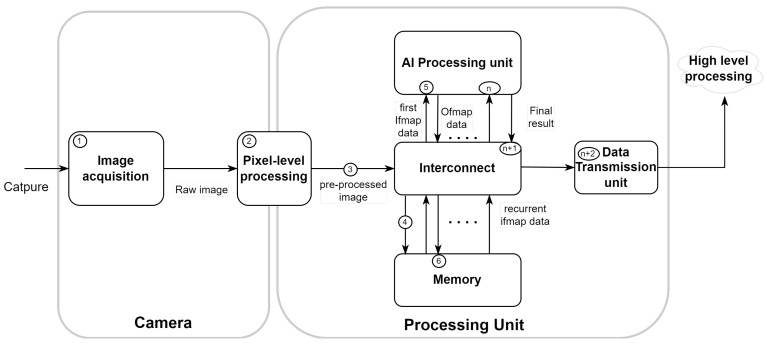
Functional view of an integrated vision system with AI processing such as [23,24,25].

**Figure 2 sensors-24-05446-f002:**
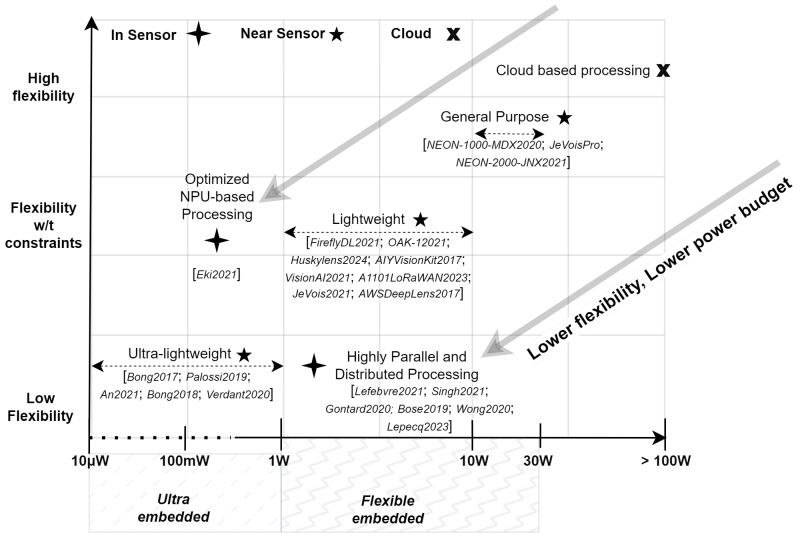
Design space for AI vision systems, highlighting the trade-off between flexibility and power consumption across processing configurations [23,29,31,32,33,34,35,36,37,38,39,40,41,42,43,44,45,46,47,48,49,50,51].

**Figure 3 sensors-24-05446-f003:**
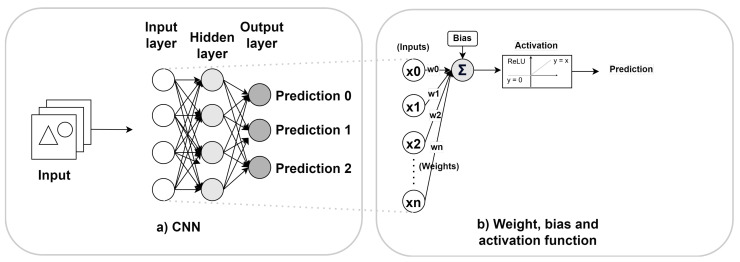
A basic CNN architecture. (**a**) Shows the network’s structure from input to output, visualizing data processing for predictions. (**b**) Details the mathematical operations, including weights, biases, and activation functions, that map inputs into outputs.

**Figure 5 sensors-24-05446-f005:**
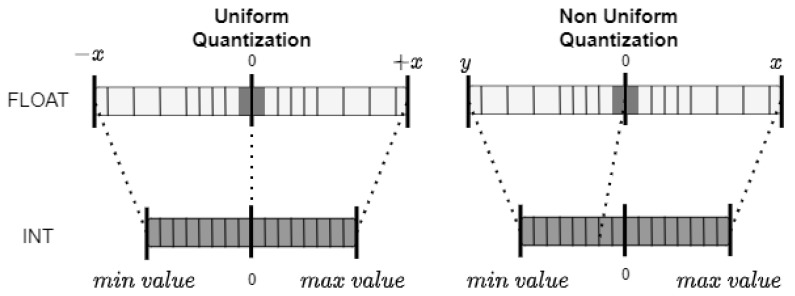
Representation of uniform and non-uniform quantization (inspired by [111]). The process converts full-precision weights and activations to lower bit-width representations, reducing the model size and computational requirements while preserving accuracy.

**Figure 6 sensors-24-05446-f006:**
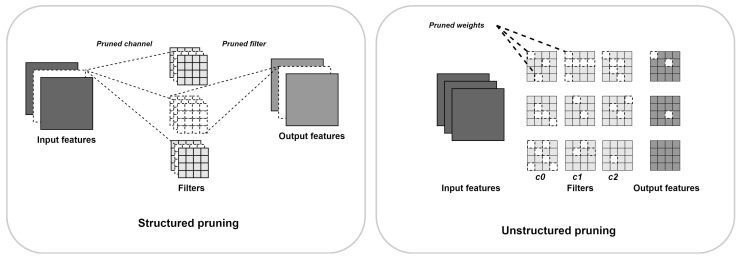
Representation of structured and unstructured pruning. Unstructured pruning removes individual weights across filters, allowing for fine-grained sparsity but potentially irregular computation patterns. Structured pruning removes entire channels or filters, resulting in a more regular pruned architecture that can be more efficiently implemented in hardware.

**Figure 7 sensors-24-05446-f007:**
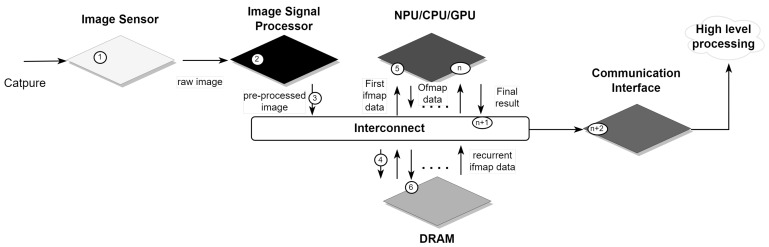
Simplified architectural view of near-sensor AI vision systems.

**Figure 8 sensors-24-05446-f008:**
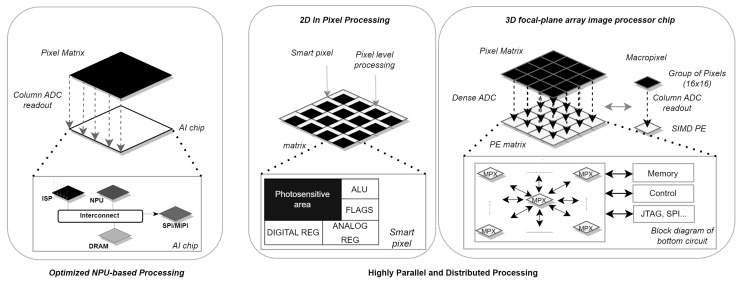
Simplified architectural view of in-sensor AI vision systems.

**Table 1 sensors-24-05446-t001:** Comparison of selected general purpose AI processing systems: This table presents three general-purpose AI processing systems, highlighting their respective processing units, software, frames per second (FPS), power consumption (in Watts), and size (in cubic centimeters). The data are based on the classification of ResNet50 FP32 from MLPerf [120]. The systems presented in the table vary in terms of their capabilities, energy efficiency, and size, demonstrating the diversity in the design space for general-purpose AI vision systems.

References	Processing	Software	FPS	Power (Watt)	Size (cm^3^)	TOPS
NVIDIA	NVIDIA Jetson AGX Orin	TFLite v2.7.1	20.1	32.46	∼500	>150
Jetson						
AGX Orin [120]						
NVIDIA	NVIDIA Jetson AGX Xavier	TFLite v2.5.0	3.8	9.45	∼50	32
Jetson						
AGX Xavier [29]						
Khadas VIM4 [34]	ARM Mali G52MP8(8EE)	ArmNN	6.7	7.64	∼50	5
		v22.05				
		(OpenCL)				

**Table 2 sensors-24-05446-t002:** Comparison of selected lightweight AI processing systems: This table presents several lightweight AI processing systems, highlighting software, frames per second (FPS), power consumption (in Watts), size (in cubic centimeters), and pre-processing techniques.

References	Processing	CNN and Software	FPS	Power (Watt)	Size (cm^3^)	Pre- Processing
[39]	Huskylens	K210 CNN	-	1.1	≈ 52.6	-
[35]	Myriad 2	MobileNet V1 1.0 224 Spinnaker SDK	12.3	1	10.6	scaling, normalization
[40]	AIY Vision Kit v1.1	constrained TensorFlow models	-	1	267.3	-
[41]	VisionAI v1.2	TFLite models	-	0.5 to 2	100.25	-
[42]	SenseCAP A1101	OD, OC, IC	-	1 to 4	-	-
[36]	Myriad X	Intel OpenVINO model zoo	-	2.5 to 5	54.5	cropping, scaling
[43]	Allwinner A33	Fine-tuned MobileNetV1_0.5 12/18 layers	7.6	4	1.65	resizing, normalization
[44]	AWS DeepLens	Apache MXNet	-	8	554.3	-

OD: object detection, OC: object counting, IC: image classification.

**Table 3 sensors-24-05446-t003:** Comparison of selected ultra-lightweight embedded AI vision systems: This table presents several ultra-embedded AI processing systems, highlighting their respective processing units, software, frames per second (FPSs), power consumption (mW), and size.

References	Processing	Software	FPSs	Power (mW)	Size	Function and Application Scenarios
[33]	PULP GAP8 (RISC V core)	Prog	6	64	-	Navigation, decision-making
[45]	Neural Engine (specialized)	Prog	PD: 5 fps	0.17	4.8 mm × 5.6 mm	PD, FD, FR
			FD: 0.28 fps			
			FR: 0.16 fps			
[32,46]	CNNP (specialized)	Configurable	1	0.62	FIS:3300 × 3300 μm, CNNP: 4 k × 4 k μm	Facial recognition

PD: person detection, FD: face detection, FR: facial recognition. FIS: face image sensor, CNNP: CNN processor, Prog: programmable.

**Table 4 sensors-24-05446-t004:** NPU-based In-Sensor AI vision systems.

References	Processing	Software	FPSs	Power (mW)	Size	Function and Application Scenarios
[31]	NPU (3D stack)	Programmable	30/120	278.7/379.1	7.558 mm × 8.206 mm	High-density, low-power CNN processing

**Table 5 sensors-24-05446-t005:** SCAMP-based In-Sensor AI Vision Systems.

References	Processing	Software	FPSs	Power (mW)	Size	Function and Application Scenarios
[139]	SCAMP-5	Configurable	210	2000	35 mm × 25 mm	In-sensor convolution, edge detection
[50]	SCAMP-5	Configurable	2260	-	35 mm × 25 mm	Fast in-sensor inference

**Table 6 sensors-24-05446-t006:** Macropixel-based in-sensor AI vision systems.

References	Processing	Software	FPSs	Power (mW)	Size	Function and Application Scenarios
[23]	MPX (3D stack)	Programmable	265	-	10 mm × 10 mm	In-sensor parallel CNN processing

## Data Availability

Data are contained within the article.

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
