# Peer review of "From Near-Sensor to In-Sensor: A State-of-the-Art Review of Embedded AI Vision Systems"

_sensors, 2024, doi:10.3390/s24165446_

Round 1

Reviewer 1 Report

Comments and Suggestions for Authors

This is a review article where the author compares and collects information on the utilization of AI in embedded vision systems. While the body of the paper adequately presents and discusses information from various sources, the introduction and conclusion have notable shortcomings.

The abstract needs to be reformulated as it is currently written like an introduction and does not emphasize the added value of this paper. The current abstract should be incorporated into the introduction as it clarifies some of its parts. 

Please find below my comments:

  1. Abstract: The abstract needs to be reformulated to highlight the added value of this paper, rather than serving as an introduction. The current abstract should be moved to the introduction to clarify some of its parts.

  2. Introduction: The introduction is fragmented and requires significant revisions:

    • The context is weak and needs to be developed and enriched with a robust background.
    • The author focuses on CNN without providing alternatives in the state of the art. While methods like MLP are mentioned, they lack in-depth comparison. Although CNN is discussed in a dedicated section, the introduction does not justify the decision to utilize it satisfactorily.
    • The objectives of the paper are fragmented across line 42, line 61, and the end of the introduction, each discussing different aspects.
    • The scope of the discussion is unclear, with various items discussed randomly and lacking connection. For instance, sections 1.1 and 1.2 discuss power consumption, though this was not stated as an important aspect earlier in the introduction.
    • The section titles should be reconsidered, as they are lengthy and limit the arguments that should be discussed within the sections themselves.
  3. Conclusion: The conclusion is lengthy and details many aspects that would benefit from being discussed in a dedicated section, such as perspectives.

Due to the issues mentioned above, I have selected major revisions for this paper.

Author Response

Manuscript ID: sensors- sensors-3134939
Response to reviewers' Comments

We are thankful to the reviewers for their thorough evaluation and valuable feedback on our manuscript. Their insightful comments and constructive suggestions have significantly contributed to the improvement of our work. By addressing each point raised, we have refined our paper, clarify our objectives, and improve the overall coherence of our discussion. We have highlighted all changes in teal within the manuscript for easy identification. A detailed, point-by-point response to each of the reviewers' comments is provided below. We are confident that the revisions have enhanced the quality of our manuscript, and we sincerely appreciate the reviewer’s expertise and guidance in this process.

Response to reviewer #1:

This is a review article where the author compares and collects information on the utilization of AI in embedded vision systems. While the body of the paper adequately presents and discusses information from various sources, the introduction and conclusion have notable shortcomings.

The abstract needs to be reformulated as it is currently written like an introduction and does not emphasize the added value of this paper. The current abstract should be incorporated into the introduction as it clarifies some of its parts. 

Please find below my comments:

  1. Abstract: The abstract needs to be reformulated to highlight the added value of this paper, rather than serving as an introduction. The current abstract should be moved to the introduction to clarify some of its parts.

Answer : As suggested by the reviewer, the abstract has been merged with the introduction to preserve key elements that introduce concepts utilized in the article. The abstract has been then rewritten to emphasize the proposed contributions: a description of the characteristics and metrics of vision systems integrating AI processing, near-sensor systems, in-sensor systems, and future perspectives (line 1 to line 21).

  1. Introduction: The introduction is fragmented and requires significant revisions:
    • The context is weak and needs to be developed and enriched with a robust background.

Answer : To enrich the background of the article, descriptive elements of pre-AI processing have been added to emphasize the use of AI processing (lines 41 to 50).

    • The author focuses on CNN without providing alternatives in the state of the art. While methods like MLP are mentioned, they lack in-depth comparison. Although CNN is discussed in a dedicated section, the introduction does not justify the decision to utilize it satisfactorily.

Answer : As suggested by the reviewer, further clarifications have been included to better highlight the unique use of CNNs in the article (lines 56 to 69). The focus is placed on the computational complexity of transformer-based processing and the substantial computational demands required to achieve high performance in MLP models.

    • The objectives of the paper are fragmented across line 42, line 61, and the end of the introduction, each discussing different aspects.

Answer : To clarify the content and as suggested by the reviewer, the objectives have been consolidated in lines 98 to 101 to clearly define the study on embedded vision systems incorporating CNN processing in the visible spectrum.

    • The scope of the discussion is unclear, with various items discussed randomly and lacking connection. For instance, sections 1.1 and 1.2 discuss power consumption, though this was not stated as an important aspect earlier in the introduction.

Answer : To facilitate a clearer discussion and to highlight important metrics that are later reused in several diagrams, we have, following the reviewer's comments, included an explicit mention of metrics such as power consumption, latency, and performance in lines 98 to 101. This emphasizes their significance as key metrics before they are applied throughout the study.

    • The section titles should be reconsidered, as they are lengthy and limit the arguments that should be discussed within the sections themselves.

Answer : The titles have been revised and shortened for clearer meaning at lines 179, 184, 207, 238, 247, and 271.

Conclusion: The conclusion is lengthy and details many aspects that would benefit from being discussed in a dedicated section, such as perspectives.

Answer : As suggested by the reviewer, the conclusion has been split into two sections: a Conclusion section, followed by a Perspectives section. The Conclusion is located at line 1192 and the Perspectives at line 1253.

Due to the issues mentioned above, I have selected major revisions for this paper.

Reviewer 2 Report

Comments and Suggestions for Authors

This manuscript reviews embedded vision systems with convolutional neural networks, highlighting trade-offs in two main approaches: near-sensor processing and in-sensor processing.

The manuscript lacks clear logic and a cohesive structure, making it difficult to follow. I recommend 'Reconsider after major revision'.

Several comments are as follows:

1. There are many instances of repetition in the manuscript, such as the first four sentences of the abstract and the first paragraph of the introduction.

2. In Figure 1, there is an intersection between two arrows. Is there a relationship between these arrows? If not, consider using a 'cross-over' to express this.

3. From Figure 2, it seems the difference between ultra embedded and flexible embedded systems is the energy budget. Please clarify this in section 1.1.

4. The formatting throughout the manuscript is inconsistent, sometimes aligning text to the left and sometimes justifying it.

5. The logic in section 2 is unclear. You attempt to describe many aspects of embedded AI vision systems, but the connection between each subsection is weak. For example, in section 2.3, you can describe from both hardware and software perspectives. However, in subsection 2.3.1, you describe hardware, and subsequent subsections discuss from different perspectives.

6. Figure 8 is unclear and does not accurately depict the relationship between the camera module and the processing unit.

7. Lines 487-489 state, "We categorize these systems into three sub-categories, as shown in Figure 8, each designed to meet distinct operational requirements." However, I do not see the three sub-categories in Figure 8.

8. The manuscript's layout is inconsistent. Some subsections are numbered while others are not. For example, in section 4.2, I recommend you write in this way ‘4.2.1 CNN Inference Process’ and 4.2.2 Opportunities for Improvement.

Comments on the Quality of English Language

1. Line 72, "AAs such," appears to be a typo.

Author Response

Manuscript ID: sensors- sensors-3134939
Response to reviewers' Comments

We are thankful to the reviewers for their thorough evaluation and valuable feedback on our manuscript. Their insightful comments and constructive suggestions have significantly contributed to the improvement of our work. By addressing each point raised, we have refined our paper, clarify our objectives, and improve the overall coherence of our discussion. We have highlighted all changes in teal within the manuscript for easy identification. A detailed, point-by-point response to each of the reviewers' comments is provided below. We are confident that the revisions have enhanced the quality of our manuscript, and we sincerely appreciate the reviewer’s expertise and guidance in this process.

Response to reviewer #2:

This manuscript reviews embedded vision systems with convolutional neural networks, highlighting trade-offs in two main approaches: near-sensor processing and in-sensor processing.

The manuscript lacks clear logic and a cohesive structure, making it difficult to follow. I recommend 'Reconsider after major revision'.

Several comments are as follows:

  1. There are many instances of repetition in the manuscript, such as the first four sentences of the abstract and the first paragraph of the introduction.

Answer: The abstract has been merged with the introduction to preserve key elements that introduce concepts utilized in the article. The abstract has been then rewritten to emphasize the proposed contributions: a description of the characteristics and metrics of vision systems integrating AI processing, near-sensor systems, in-sensor systems, and future perspectives (line 1 to line 21).

  1. In Figure 1, there is an intersection between two arrows. Is there a relationship between these arrows? If not, consider using a 'cross-over' to express this.

Answer: Figure 1 has been revised to not only clarify the illustration and eliminate the issue of crossover on the lines but also to enhance the clarity of the content and better integrate the figure into the paper (page 3).

  1. From Figure 2, it seems the difference between ultra embedded and flexible embedded systems is the energy budget. Please clarify this in section 1.1.

Answer: To clarify the content and as suggested by the reviewer, the objectives have been consolidated in lines 98 to 101 to clearly define the study on embedded vision systems incorporating CNN processing in the visible spectrum.

  1. The formatting throughout the manuscript is inconsistent, sometimes aligning text to the left and sometimes justifying it.

Answer: The titles have been revised and shortened for clearer meaning at lines 179, 184, 207, 238, 247, and 271. There is no more utilization of the latex command \paragraph{} that changes between aligning text to the left and sometimes justifying it.

  1. The logic in section 2 is unclear. You attempt to describe many aspects of embedded AI vision systems, but the connection between each subsection is weak. For example, in section 2.3, you can describe from both hardware and software perspectives. However, in subsection 2.3.1, you describe hardware, and subsequent subsections discuss from different perspectives.

Answer: As suggested by the reviewer, we have clarified Section 2 (page 6 to 18). by providing a detailed description of the characteristics and metrics from the image capture to the end of the CNN processing. This matches the exact sequence illustrated in Figure 1, which serves to support this explanation.

  1. Figure 8 is unclear and does not accurately depict the relationship between the camera module and the processing unit.

Answer:  As suggested by the reviewer, we have enhanced Figure 8 on page 24 by adding a detailed description of the composition of each key element in every in-sensor vision system. For optimized NPU-based processing, we included a clear representation of the AI chip. For 2D in-pixel processing, we added a description of a smart pixel, and for the 3D focal-plane array, we provided a pictorial representation of the bottom processing block.

  1. Lines 487-489 state, "We categorize these systems into three sub-categories, as shown in Figure 8, each designed to meet distinct operational requirements." However, I do not see the three sub-categories in Figure 8.

Answer: To better illustrate the categories, we added clear separations, and this type of separation has also been incorporated into all diagrams to ensure greater consistency throughout the paper.

  1. The manuscript's layout is inconsistent. Some subsections are numbered while others are not. For example, in section 4.2, I recommend you write in this way ‘4.2.1 CNN Inference Process’ and 4.2.2 Opportunities for Improvement.

Answer: All subsections and sub-subsections have been revised. As recommended by the reviewer, we have organized the “CNN Inference Process” and “Opportunities for Improvement” as sub-subsections to enhance the paper's readability.

Comments on the Quality of English Language

  1. Line 72, "AAs such," appears to be a typo.

Answer: Thank you for pointing out this typo. We have corrected "AAs such" to "As such" on line 72.

Reviewer 3 Report

Comments and Suggestions for Authors

This manuscript provides a broad overview of state-of-the-art near-sensor and in-sensor AI vision systems. The sections are overall well organized, and the subject matter is highly relevant to the embedded AI space. For a more holistic survey, I believe that the manuscript can benefit from several additional discussions as detailed below:

1. Since this submission focuses on providing a survey, I recommend that the authors expand Tables 1, 2, 3 to include more designs for comparison.

2. For the near-sensor section, discussion on 2.5-D integration should be included. For instance, technologies such as the interposer and system-in-package are relevant to this field, and will be crucial to physically couple processing and sensing. Some industry and academic examples may be added to illustrate.

3. A major advantage of near-sensor and in-sensor systems is to perform some processing close to where the raw data is generated, in order to alleviate the system bandwidth and eventually reduce the amount of data that the backend processors have to deal with. Please include some quantitative discussions on how some of the surveyed designs can help with the bandwidth.

4. On the topic of data movement, the interconnect between the image sensor and other components like processor and memory will be important to discuss. For example, what are some typical interconnects and protocols that near-sensor and in-sensor systems utilize? How do they compare to traditional AI vision where all the raw data are sent to the backend?

Author Response

Manuscript ID: sensors- sensors-3134939
Response to reviewers' Comments

We are thankful to the reviewers for their thorough evaluation and valuable feedback on our manuscript. Their insightful comments and constructive suggestions have significantly contributed to the improvement of our work. By addressing each point raised, we have refined our paper, clarify our objectives, and improve the overall coherence of our discussion. We have highlighted all changes in teal within the manuscript for easy identification. A detailed, point-by-point response to each of the reviewers' comments is provided below. We are confident that the revisions have enhanced the quality of our manuscript, and we sincerely appreciate the reviewer’s expertise and guidance in this process.

Response to reviewer #3:

This manuscript provides a broad overview of state-of-the-art near-sensor and in-sensor AI vision systems. The sections are overall well organized, and the subject matter is highly relevant to the embedded AI space. For a more holistic survey, I believe that the manuscript can benefit from several additional discussions as detailed below:

  1. Since this submission focuses on providing a survey, I recommend that the authors expand Tables 1, 2, 3 to include more designs for comparison.

Answer: To create a more comprehensive survey, I made an effort to identify new vision systems that integrate AI processing both near-sensor and in-sensor. As a result, I discovered several new vision systems: HuskyLens, AIY Vision Kit v1.1, SenseCAP A1101, and AWS DeepLens, all of which belong to the lightweight category (line 764). Despite my efforts, I was unable to find additional systems, particularly those in the in-sensor category.

  1. For the near-sensor section, discussion on 2.5-D integration should be included. For instance, technologies such as the interposer and system-in-package are relevant to this field, and will be crucial to physically couple processing and sensing. Some industry and academic examples may be added to illustrate.

Answer: The discussion on 2.5D systems is indeed very interesting, which is why it has been included in the perspective section (line 1282 to 1304). There was not enough information available on vision systems integrating AI in 2.5D. These systems are particularly intriguing because they allow for heterogeneous integration and fall somewhat between the categories of near-sensor and in-sensor processing. They closely resemble the 3D system proposed by SONY in the category of Optimized NPU-based processing.

  1. A major advantage of near-sensor and in-sensor systems is to perform some processing close to where the raw data is generated, in order to alleviate the system bandwidth and eventually reduce the amount of data that the backend processors have to deal with. Please include some quantitative discussions on how some of the surveyed designs can help with the bandwidth.
  2. On the topic of data movement, the interconnect between the image sensor and other components like processor and memory will be important to discuss. For example, what are some typical interconnects and protocols that near-sensor and in-sensor systems utilize? How do they compare to traditional AI vision where all the raw data are sent to the backend?

Answer: These comments were particularly important to us, so we decided to not only add a discussion on bandwidth but also link it to a discussion on interconnects at the same time. First, we introduced the need to reduce the amount of data through pre-processing via the ISP, including an example of binning (line ..). We then discussed bandwidth as a major factor to consider, identifying two key types: the bandwidth between image capture and pixel pre-processing (analog to digital) and the bandwidth between digital elements enabling CNN inference.

This discussion allowed us to highlight the different bandwidth requirements based on near-sensor and in-sensor systems, with a clear distinction between the different types of pixel readouts. By examining communication between digital elements, we emphasized various bus types and protocols, along with their trade-offs, bandwidth capabilities, and integration potential. These factors include size, power consumption, and whether the bus is external, internal, serial, or parallel.

(line 224 to 302)

Reviewer 4 Report

Comments and Suggestions for Authors

1) In the section “2.2 Evolution of embedded convolutional neural networks”, authors example several CNN models, such as VGGNet, GoogLeNet, ResNet, and so on. At the end of the section, authors categorize the CNN models into three groups. It is better to categorize above CNN models into these three groups, to illustrate the conclusion.

2) In the section “2.3.1 System hardware …” line 281, the Memory resources. The article mentions the memory is essential for storing the CNN model. However, another more important function of memory is for storing the middle generation data and feature map data, which are massive in some applications.

3) In the section of 2.3, the subtitles from 2.3.1 to 2.3.6 are organized in confused sequence, neither the flow of data processing, nor the importance of every capabilities. It is suggested to organized as flow of data capturing and processing.

4) Figure 5 does not illustrate well what is expressed in the text and does not convey the characteristics of the quantization technique.

5) Figure 6 is not illustration well what is expressed in the text and does not convey the characteristics of the pruning technique.

6) Figure 7 is an inappropriate representation of article content, “KD loss” is not explained and showed the connection if figure 7.

7) In Table 1, it is suggested to add a “TOPS” column, to illustrate the computational capacities of the systems.

8) In Table 2, it is suggested to add a “Pre-processing” column, to illustrate the flexibility of the systems.

9) In Table 3, it is suggested to add a “Function and application scenarios” column, to illustrate the advantages of the systems.

10) In the section 4, it is suggested to add Tables to compare the characteristics of In-sensor processing AI vision systems, like Table 1~3.

Comments on the Quality of English Language

Moderate editing of English language required.

Author Response

Manuscript ID: sensors- sensors-3134939
Response to reviewers' Comments

We are thankful to the reviewers for their thorough evaluation and valuable feedback on our manuscript. Their insightful comments and constructive suggestions have significantly contributed to the improvement of our work. By addressing each point raised, we have refined our paper, clarify our objectives, and improve the overall coherence of our discussion. We have highlighted all changes in teal within the manuscript for easy identification. A detailed, point-by-point response to each of the reviewers' comments is provided below. We are confident that the revisions have enhanced the quality of our manuscript, and we sincerely appreciate the reviewer’s expertise and guidance in this process.

Response to reviewer #4:

1) In the section “2.2 Evolution of embedded convolutional neural networks”, authors example several CNN models, such as VGGNet, GoogLeNet, ResNet, and so on. At the end of the section, authors categorize the CNN models into three groups. It is better to categorize above CNN models into these three groups, to illustrate the conclusion.

 Answer: As suggested by the reviewer, we have made changes to immediately categorize the different CNNs. To implement this change, we first described the categories and classified the CNNs as we introduced them. We chose to maintain a chronological order in the descriptions because it illustrates the successive contributions of CNNs relative to one another. This approach not only highlights the chronological progression but also allows us to trace the evolution of ideas and concepts that lead to the development of the most advanced CNNs for embedded systems (lines 360 to 498).

2) In the section “2.3.1 System hardware …” line 281, the Memory resources. The article mentions the memory is essential for storing the CNN model. However, another more important function of memory is for storing the middle generation data and feature map data, which are massive in some applications.

 Answer: The reviewer highlighted an essential point that was not addressed in our study. In response, we corrected this oversight by emphasizing the storage and bandwidth requirements of CNNs for partial sums and feature maps (lines 224 to 328).

3) In the section of 2.3, the subtitles from 2.3.1 to 2.3.6 are organized in confused sequence, neither the flow of data processing, nor the importance of every capabilities. It is suggested to organized as flow of data capturing and processing.

 Answer: As suggested by the reviewer, and to expand upon this particularly important comment, we have restructured the entire Section 2, as well as Figure 1, to improve the readability and understanding of the characteristics and metrics of vision systems integrating AI. Figure 1 has been revised to illustrate the entire process, from image capture to the end of inference. This reorganization allows Section 2 to rely on the figure to highlight the important characteristics and metrics throughout the process. We also added numerous transitions between the existing paragraphs, which have been repositioned for better coherence.
(Page 6 to 18)

4) Figure 5 does not illustrate well what is expressed in the text and does not convey the characteristics of the quantization technique.

 Answer: To enhance clarity, we have modified Figure 5 to illustrate the process of uniform and non-uniform quantization, demonstrating the conversion from floating-point numbers (FLOATS) to integers (INTS). This revision provides a clearer visual representation of the quantization techniques discussed in the text, making it easier for readers to understand the distinctions and applications of these methods. (Page 14)

5) Figure 6 is not illustration well what is expressed in the text and does not convey the characteristics of the pruning technique.

 Answer: We have revised Figure 6 to better illustrate structured and unstructured pruning, as discussed in the text. This updated figure highlights the significance of pruning by visually demonstrating the benefits of pruning channels, filters, and weights. It provides readers with a clearer understanding of how these pruning techniques contribute to optimizing neural networks, reinforcing the key concepts presented in our paper. (Page 15)

6) Figure 7 is an inappropriate representation of article content, “KD loss” is not explained and showed the connection if figure 7.

 Answer: As the reviewer correctly pointed out, Figure 7 was not well correlated with the rest of the paper. Consequently, we have removed the figure and its accompanying text to maintain the paper's focus and coherence (page ).

7) In Table 1, it is suggested to add a “TOPS” column to illustrate the computational capacities of the systems.

Answer: We have added a "TOPS" column to Table 1, which now clearly illustrates the computational capacities of the systems. This addition provides a more comprehensive understanding of the performance metrics and helps readers easily compare the systems based on their throughput.

8) In Table 2, it is suggested to add a “Pre-processing” column to illustrate the flexibility of the systems.

Answer: A "Pre-processing" column has been added to Table 2, highlighting the flexibility of the systems regarding their ability to handle various pre-processing tasks. This enhancement allows readers to better appreciate the systems' adaptability and versatility.

9) In Table 3, it is suggested to add a “Function and application scenarios” column to illustrate the advantages of the systems.

Answer: We have included a "Function and Application Scenarios" column in Table 3. This new column provides detailed insights into the practical applications and benefits of each system, offering a more comprehensive view of their advantages and potential use cases.

10) In Section 4, it is suggested to add tables to compare the characteristics of in-sensor processing AI vision systems, like Table 1~3.

Answer: In Section 4, we have added several tables that compare the characteristics of in-sensor processing AI vision systems, similar to the format of Tables 1-3. These tables provide a structured comparison of key features, allowing for a clearer understanding of the differences and advantages of each system.

Round 2

Reviewer 1 Report

Comments and Suggestions for Authors

Author have taken into consideration all my previous comments.

Reviewer 2 Report

Comments and Suggestions for Authors

The changes and revisions made by the authors are much appreciated. I recommend to publish this manuscript in Sensors.

Reviewer 3 Report

Comments and Suggestions for Authors

I would like thank the authors for addressing my comments and taking the efforts to improve this manuscript. I have no further questions.

Reviewer 4 Report

Comments and Suggestions for Authors

The authors have addressed all of my concerns. I would like to thank them for the details.

Comments on the Quality of English Language

Minor editing of English language required.